**Investigation**

# Chimeric reference panels for genomic imputation

Meikun Zhou 🔾,* Maddie E. James 🔾, Jan Engelstädter, Daniel Ortiz-Barrientos 🔾

School of the Environment, and Australian Research Council Centre of Excellence for Plant Success in Nature and Agriculture, The University of Queensland, St Lucia, QLD 4072, Australia

*Corresponding author: School of the Environment, and Australian Research Council Centre of Excellence for Plant Success in Nature and Agriculture, The University of Queensland, St Lucia, QLD, 4072, Australia. Email: meikun.zhou@uq.edu.au

Despite transformative advances in genomic technologies, missing data remain a fundamental constraint that limits the full potential of genomic research across biological systems. Genotype imputation offers a remedy by inferring unobserved genotypes from observed data. However, conventional imputation methods typically rely on external reference panels constructed from complete genome sequences of hundreds of individuals, a costly approach largely inaccessible for nonmodel organisms. Moreover, these methods generally overlook novel genomic positions not captured in existing panels. To overcome these limitations, we developed Retriever, a method for constructing a chimeric reference panel that enables genotype imputation without the need for an external reference panel. Retriever constructs a chimeric reference panel directly from the target samples using a sliding window approach to identify and retrieve genomic partitions with complete data. By exploiting the complementary distribution of missing data across samples, Retriever assembles a panel that preserves local patterns of linkage disequilibrium and captures novel variants. When the Retriever-constructed panels are used with Beagle for genotype imputation, Retriever consistently achieves accuracy exceeding 95% across diverse datasets, including plants, animals, and fungi. By eliminating the need for costly external panels, Retriever provides an accessible and cost-effective solution that broadens the application of genomic analyses across various species.

Keywords: imputation; reference panel; nonmodel organisms; missing data; next-generation sequencing; VCF file

## Introduction

Genomic research has transformed our understanding of biological systems, from elucidating phenotype–genotype associations (Visscher et al. 2017) to uncovering mechanisms of adaptation (Barrett and Hoekstra 2011). This transformation is fueled by technological advances: next-generation sequencing now generates over 100 gigabases per run (Goodwin et al. 2016), while long-read technologies produce continuous sequences exceeding 100 kilobases (Logsdon et al. 2020). Despite these advances, missing genomic data persist as a fundamental constraint. Datasets routinely contain substantial gaps in genotypes. This can produce inaccurate population diversity and differentiation statistics (Schmidt et al. 2021; Bailey et al. 2025), distort the construction of linkage maps (Hackett and Broadfoot 2003), and reduce power and increase false discovery rates for studies of rare variants (Auer et al. 2013).

Genotype imputation addresses these challenges by inferring unobserved genotypes through linkage disequilibrium patterns and haplotype structure (Li et al. 2009). Methods employing hidden Markov models, such as Beagle (Browning and Browning 2016), IMPUTE2 (Howie BN et al. 2009), and MaCH (Li et al. 2010) demonstrate exceptional effectiveness. The impact is substantial: human genomics studies have expanded from thousands of directly genotyped variants to millions of imputed variants in projects like the UK Biobank (Bycroft et al. 2018). Imputation has significantly enhanced genomic prediction accuracy for economically important agricultural traits (Gorjanc et al. 2017), enabled

more cost-effective breeding strategies (Tsai et al. 2017), and shows growing potential to strengthen conservation genomics efforts (Theissinger et al. 2023) and ecological research (Taş et al. 2021).

However, conventional imputation frameworks such as Beagle, IMPUTE2, and MaCH, share a critical limitation: dependence on external reference panels comprising hundreds of fully genotyped individuals that represent the genetic diversity of the target population (Das et al. 2016; Phocas 2022b; Dekeyser et al. 2023). These methods assume that missing genotypes in target samples follow predictable patterns that can be inferred from complete reference data (Li et al. 2009; Browning and Browning 2016), but face fundamental limitations when reference panels are unavailable or incomplete (Marchini and Howie 2010; Das et al. 2016). This requirement creates a methodological divide in genomic research: while millions of human genomes have been sequenced, very few species possess external reference panels suitable for conventional imputation. Panel construction requires substantial resources, specialized expertise, and infrastructure that are typically available only for major model organisms. Additionally, conventional imputation approaches ignore and disregard any novel genomic positions in the target samples that are absent from the reference panel (Marchini and Howie 2010; Huang GH and Tseng 2014). This problem intensifies as sequencing technologies advance and capture novel variants with potential functional importance (Satam et al. 2023), potentially losing valuable biological information (Marchini and Howie 2010; Huang GH and Tseng 2014).

Here, we introduce Retriever, a framework designed to make conventional imputation algorithms—which typically require an external reference panel—accessible to nonmodel organisms. Retriever enables the use of conventional imputation methods by creating a chimeric reference panel directly from the target dataset. Our approach uses sliding windows to identify genomic partitions with complete data across multiple samples. By maintaining local patterns of linkage disequilibrium within these windows, the algorithm preserves the haplotype structure critical for accurate imputation. The chimeric reference panel effectively reconfigures the patchy landscape of missing data by leveraging the observation that, while any single sample may have missing genotypes, the collective dataset often contains complete information at each position across different subsets of individuals.

We use rigorous benchmarking to demonstrate that Retriever, when used in combination with Beagle for imputation, maintains >95% imputation accuracy across allele frequency spectra in datasets without an external reference panel. Comparative analyses against both reference-based and reference-free imputation methods confirm that our chimeric panel strategy combines the accessibility of reference-free approaches with the precision of reference-based algorithms by minimizing disruption to underlying linkage patterns. Retriever thus makes previously inaccessible imputation algorithms available for nonmodel organisms, enabling researchers to extract maximum information from genomic resources and potentially reveal novel patterns that missing data would otherwise obscure.

## Materials and methods
### Input files and requirements
Retriever operates on Variant Call Format (VCF) files v4.0 or higher. We use the term "target samples" to refer to the collection of individuals that contain missing genotypes, which require imputation. The algorithm assumes that observed genotypes in the VCF file are error-free. Therefore, users must have previously implemented comprehensive quality control procedures, including genotype filtering based on quality metrics and read depth (as outlined by O'Leary et al. 2018). This preprocessing should include the removal of multiallelic sites, which can complicate haplotype inference, and the elimination of positions with greater than 20% missing data.

To clarify the distinction between preprocessing requirements and our testing parameters: the requirement to remove positions with >20% missing data refers to genomic positions where more than 20% of individuals have missing genotypes, which can cause computational instability during chimeric panel assembly. In contrast, our testing with up to 30% missing data (as described below) involves randomly distributing missing genotypes across the entire dataset, while maintaining the requirement that no single position exceeds 20% missingness. This approach ensures computational stability while demonstrating Retriever's robustness under challenging scenarios of overall missingness that may occur in real datasets.

As Retriever uses a sliding window approach, it should be executed on a per-chromosome or contig basis. Importantly, Retriever must be implemented on cohorts of genetically similar samples, such as individuals within a population, ecotype, or species, as substantial genetic divergence is expected to compromise the underlying assumption of shared haplotype structure. GVCF files, which contain additional reference blocks for nonvariant regions, are not encouraged for use in Retriever. The implementation of Retriever uses established Python packages for processing and manipulating VCF files. A list of external dependencies is documented in the project's GitHub repository (https://github.com/uqmzhou8/Retriever.git), and a YAML configuration file is available to facilitate the seamless installation of these dependencies.

## Assembly of the chimeric reference panel
The central innovation of Retriever is the construction of a chimeric reference panel from the target data itself, thereby eliminating the need for an external reference panel (Fig. 1). This chimeric assembly process identifies and retrieves genotype information across the genomic landscape of the population. To construct the chimeric reference panel, Retriever first parses the VCF file and implements a nonoverlapping sliding window approach with a user-defined size (default: 1,000 bp) that sequentially scans all genomic positions within the target samples. For each window, the algorithm identifies sets of complete genotype partitions—groups of individuals with no missing genotypes within the current window—and stores these partitions in separate data structures, referred to as "buckets" (note that missing genotypes are encoded as −3 in our implementation to distinguish them from valid genotype calls). Before storage, the algorithm verifies that the number of individuals with complete genotype data in the window meets the user-defined minimum number of chimeric individuals to be generated. If this requirement is not satisfied, the window shrinks by 1 bp increments until complete partitions with the minimum number of individuals are identified. Should this condition remain unsatisfied despite window contraction, potentially due to inadequate filtering of the initial dataset, the algorithm will prematurely terminate and produce an error message.

As the sliding window progresses through the genome, the window size is reset at each new position. Upon completing the genome scan, the algorithm assembles the chimeric reference panel by chronologically concatenating the genotype partitions stored in the respective buckets. In cases where a bucket contains more individuals than the requested chimeric reference size, individuals are randomly selected to maintain a consistent panel size across the genome. While this approach ensures compatibility with imputation algorithms that expect fixed panel sizes, future implementations could exploit variable bucket sizes to maximize information content where sufficient complete data is available. This assembly procedure yields a chimeric reference panel comprising individuals with contiguous, nonmissing genotype data, compatible with most established imputation algorithms that require external reference panels.

## Comparison of chimeric versus external reference panels
We validated the imputation accuracy using Retriever's chimeric reference panel against an external reference panel of the same size, using a well-established imputation software as a benchmark for the highest achievable imputation accuracy. We selected Beagle4.1 (beagle.27Jan18.7e1.jar) as the imputation engine due to its widespread adoption and consistently high performance reported in the literature (Das et al. 2016). Beagle4 was preferred over its successor, Beagle5, as it was determined to be more suitable for our study after conducting several tests (Supplementary Fig. 1). While Beagle4 does require a genetic map, it provides reasonable default assumptions (uniform recombination rate of 1 cM/Mb) when a genetic map is not provided, making it more accessible than other imputation software. In contrast, other imputation software, such as EagleImp (Wienbrandt and Ellinghaus 2022), require additional resources, including both an external

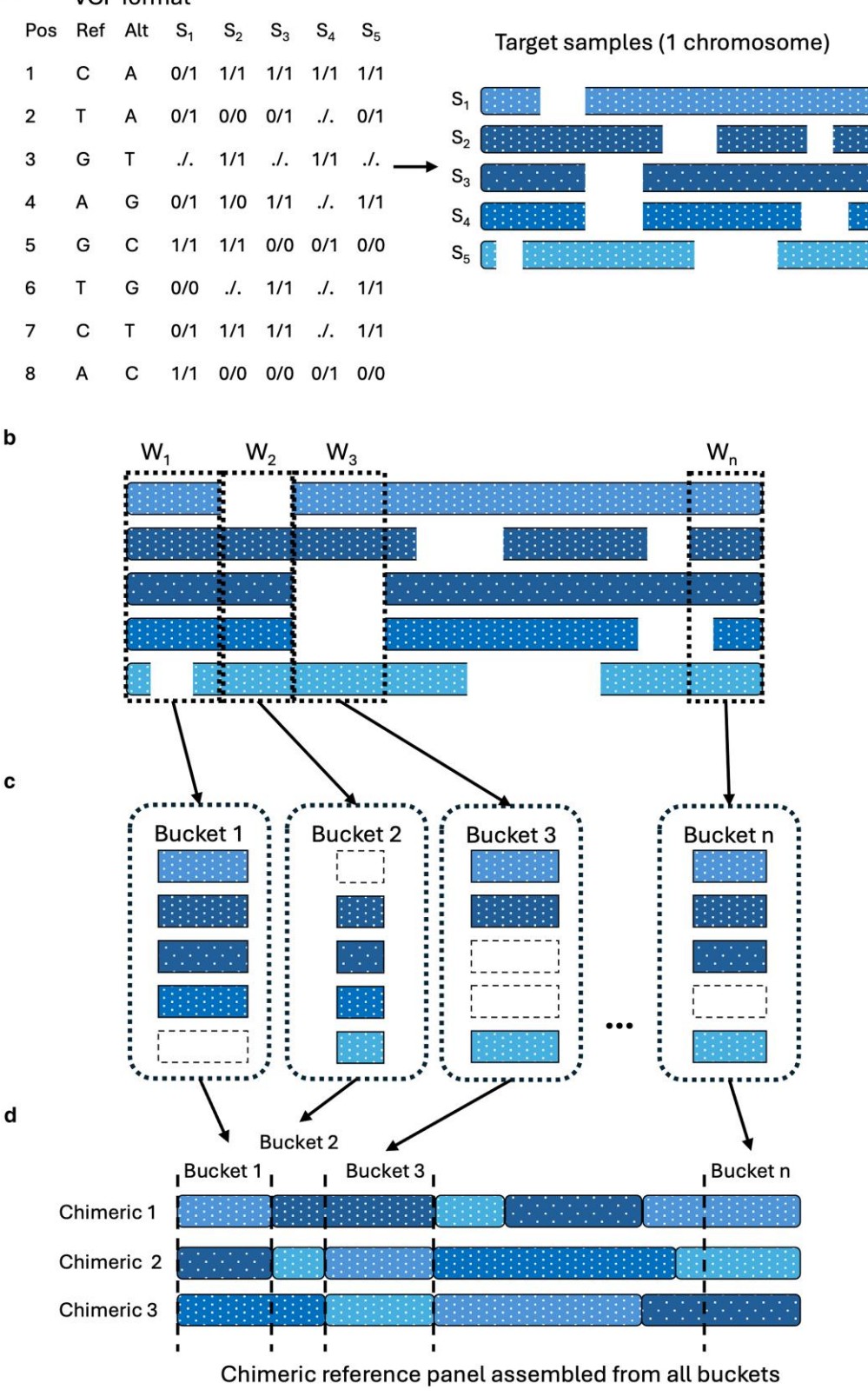

**Fig. 1.** Steps for assembly of Retriever's chimeric reference panel. a) A VCF file containing the diploid target samples ($S_1$, $S_2$, ... $S_5$) from a single population, ecotype, or species, aligned to a reference genome, is read by Retriever. The individual genomes of the five target samples are represented by the bars, and gaps represent missing genotypes. b) A nonoverlapping sliding window (W) moves across the genome, selecting complete partitions—genotypes without missing data—to be stored in buckets. If the partition of complete genotypes is less than the user-defined chimeric reference panel size, the window shrinks until the required number of partitions is reached. c) Partitions of complete genotypes are stored in individual buckets with nonoverlapping genomic positions. d) Within each bucket, individuals are randomly selected to assemble the chimeric reference panel in contiguous genomic order. In this example, the user-defined size for the chimeric reference panel is three, so three individuals are randomly selected per bucket.

reference panel of sufficient size (typically hundreds to thousands of individuals) and a detailed genetic map for accurate phasing, with no reasonable defaults when these resources are unavailable. These dual requirements—assembling a comprehensive external reference panel and constructing detailed genetic maps—can be challenging to meet, particularly for laboratories working with nonmodel organisms, where such resources are often unavailable or prohibitively expensive to generate. For instance, a forced initiation of EagleImp without a genetic map can result in an early termination or simply an error prompt upon running the program, making it unsuitable for use with Retriever's chimeric reference panel.

Each analysis (outlined below) was replicated three times by randomly masking different locations of the target samples to assess consistency and mean accuracy values and their associated standard errors. All analyses presented in this study were executed on the National Computational Infrastructure in Australia. The assembly of the chimeric reference panel was performed using a 24-core Intel Xeon Platinum 8274 (Cascade Lake) processor, while imputation with Beagle4 utilized a 24-core Intel Xeon Platinum 8,268 (Cascade Lake) processor. The imputation process utilized 96 threads with the default parameters of Beagle4 (Browning and Browning 2016) consistently across all analyses, unless otherwise stated, to ensure uniformity in our methods.

### Sample selection

To validate imputation accuracy and demonstrate Retriever's broad applicability, we used data from the human (*Homo sapiens*) 1K Genome Project (1000 Genomes Project Consortium et al. 2015) alongside genomic datasets from diverse organisms, including the plant *Arabidopsis thaliana*, the animal *Gallus gallus*, and the fungus *Saccharomyces cerevisiae*. The 1K Genomes Project was chosen due to its high quality and the availability of a large number of samples suitable for constructing an external reference panel for Beagle4. It is also a widely used benchmark in the field and has been employed in the validation of various imputation tools (Delaneau et al. 2008; Browning et al. 2018; Wienbrandt and Ellinghaus 2022).

The individuals in the current study were chosen to be of African origin, as they exhibit the highest nucleotide diversity among all human populations (Yasukochi et al. 2019). This choice allowed us to test Retriever's performance under a conservative scenario that represents a challenging condition for imputation, thereby providing a robust assessment of Retriever's capabilities. From the 661 individuals of African ancestry, we randomly selected 200 individuals as target samples for imputation. From the remaining pool, 50 individuals were randomly chosen to construct the external reference panel. This external reference panel serves as a baseline for Beagle4, representing the highest achievable imputation accuracy for a panel of similar size. In contrast, the chimeric reference panel of Retriever is assembled from the target samples. We used chromosome 1 for analysis because it is the largest chromosome, which reduces computational complexity compared to analyzing the entire genome.

### Data masking

We simulated missing genomic data by randomly masking genotypes across all individuals and genomic positions. Both alleles at each diploid site were masked to ensure that there was no data leakage for downstream analysis, as the presence of one allele that remains unmasked in a heterozygous site may potentially bias the probability of identifying the other allele. Additionally,

to examine how well the chimeric reference panel captures novel variants in the target samples, we removed random genomic positions from the external reference panel while keeping them in the target samples. This simulates real-world scenarios where target samples contain novel variants not present in pre-existing reference panels—a common occurrence when newer sequencing technologies or higher coverage reveal variants missed during external panel assembly.

### Parameter optimization

We conducted several tests to investigate the impact of chimeric panel size and window length on imputation accuracy. Theoretically, larger reference panels should yield more accurate results. However, the maximum size of the panel is constrained by the dataset's filtering thresholds, as the chimeric assembly selects the most complete haplotypes within the sliding window. Therefore, the maximum achievable size of the chimeric reference panel depends on the number of individuals with complete data at the genomic position with the highest proportion of missing genotypes. The window length determines the largest size of partitions used for constructing the chimeric panel; a larger window size is less likely to disrupt the linkage disequilibrium of the individuals. Using the human dataset, we tested a range of panel sizes (from 10 to 100 individuals) and window lengths (100 to 10,000 base pairs) across various masking proportions (1%, 10%, 20%, and 30%) to identify the optimal balance between these variables. The maximum achievable size of the chimeric reference panel is inherently constrained by the dataset's missing data patterns, as the algorithm requires complete genotypes within each sliding window. Therefore, we evaluated how different panel size requirements affected both window size dynamics and downstream imputation accuracy.

Additionally, we analyzed parameters that may affect the imputation process. Genotype imputation in Beagle4 involves two key steps: phasing (determining which alleles are inherited together on the same chromosome) and imputation of missing genotypes based on haplotype patterns. By default, Beagle4 assumes a uniform recombination rate of 1 centimorgans per megabase (cM/Mb) when a genetic map is not provided. These recombination rate assumptions are crucial for accurately determining haplotype boundaries during the phasing step, which in turn affects imputation accuracy. Since our study focuses on nonmodel organisms that typically lack detailed genetic maps with species-specific recombination rates, we sought to understand whether the default recombination rate assumption significantly affects imputation performance when using chimeric reference panels. We tested the effect of varying recombination rates on imputation accuracy by providing Beagle4 with genetic maps containing uniform recombination rates ranging from 0.5 to 2 cM/Mb, encompassing the range of recombination rates observed across diverse species.

### Statistical framework for accuracy assessment

The imputed genotypes obtained from the chimeric and external panels were compared to the original unmasked samples to determine the average imputation accuracy of all individuals at each position. To evaluate the imputation accuracy of each panel in relation to the minor allele count, an average accuracy from all three replicates was binned by the minor allele count (MAC) associated with each position. The MAC was derived from the original dataset using BCFtools v1.9 (Danecek et al. 2021). For each genomic position, we calculated accuracy as the proportion of correctly imputed genotypes among all masked genotypes. Since Beagle4

performs phasing as part of its imputation process (although the algorithm prioritizes accuracy over completeness, so it will leave sites unphased rather than make low-confidence phase calls), our method employs a stringent matching criterion: a heterozygous genotype (0|1) must be imputed exactly as 0|1 to be considered correct; an imputation of 1|0, though functionally equivalent in unphased data, is counted as incorrect because it represents a different phasing. This approach differs from conventional accuracy assessments, which focus solely on allele dosage. These assessments often encode genotypes numerically (e.g. homozygous as 1 or 3, heterozygous as 2), thereby treating 0/1 and 1/0 as identical regardless of phase. Our phase-aware accuracy metric provides a more stringent evaluation of imputation quality, taking into account the preservation of haplotype structure. We accounted for the relationship between imputation accuracy and allele count by stratifying positions by their MAC.

### Scaling of retriever

We evaluated the computational resources (wall time) required for constructing the chimeric reference panel with varying numbers of individuals from the human dataset outlined above and different proportions of masked (missing) data. Retriever was designed to process the genome using a single, continuous sliding window that spans all genomic positions within a chromosome or contig, from start to end. While this approach increases computational time compared to a partitioned approach, it preserves the integrity of linkage disequilibrium patterns in the resulting chimeric reference panel. Partitioning each chromosome into separate segments for parallel processing would create artificial boundaries between adjacent regions, potentially fragmenting haploblocks that span these boundaries. This fragmentation would increase the likelihood of combining genotypes from different individuals at these boundaries, thereby disrupting important patterns of linkage disequilibrium. The single-window approach, therefore, maintains biological accuracy while requiring modest computational resources (just 1 CPU).

### Evaluation of nonhuman organisms

We further expanded our validation to additional species that lack the resources to assemble an external reference panel and are, therefore, unable to use conventional genomic imputation software. To achieve this, we sourced publicly available datasets and selected organisms from various taxonomic kingdoms, including plants, animals, and fungi. For plants, we used *A. thaliana* sequence data from the Weigel laboratory at the Max Planck Institute for Developmental Biology (The 1001 Genomes Consortium 2016). From this genomic dataset consisting of 1,135 individuals, 200 individuals were randomly selected for the target samples. The animal kingdom was represented by 188 individuals of chicken *G. gallus* data (Cho et al. 2022). Fungi were represented by a yeast *S. cerevisiae* dataset of 165 individuals (Sardi et al. 2018). As with the human dataset, masking was applied to each set of target samples, and only chromosome 1 was used for analysis. Using the same experimental design employed for the human validation studies—including chimeric reference panel construction with optimal parameters (50 individuals), random genotype masking (1–30%), and accuracy assessment via comparison with original unmasked data—we evaluated the performance of Retriever across diverse datasets.

To validate Retriever's performance in a nonmodel organism, we analyzed 287 *Helianthus annuus* samples (Todesco et al. 2020). While *H. annuus* has some available genomic resources; these are more limited compared to extensively characterized model

organisms like humans, where large-scale consortia have generated comprehensive reference panels capturing broad population diversity. We imputed the original dataset using Beagle4 with Retriever's chimeric panel and compared the results to those obtained with an external reference panel from Todesco et al. (2020). We acknowledge that this comparison provides a useful benchmark rather than a definitive assessment of the ground truth. To evaluate the impact of imputation on downstream population genetic analyses, we inferred recombination rates using Pyrho v0.1.6 (Evans et al. 2016) for both the original dataset containing missing genotypes and the Retriever-imputed dataset, comparing these results with those reported by Todesco et al. (2020). Pyrho analysis was run with a block penalty of 100, a window size of 100 SNPs, and with 48 threads. The analysis was conducted on chromosome 1 with a minimum minor allele frequency threshold ≥3% to ensure robust rate estimation.

### Benchmarking with reference-free imputation

Using the human dataset described above, we conducted benchmarking against other commonly used reference-free imputation algorithms: k-nearest neighbors (KNN) (Troyanskaya et al. 2001), and missForest (Stekhoven and Bühlmann 2012). For KNN-based imputation, we utilized the implementation available in the Python Scikit-learn package v1.6.1 (Pedregosa et al. 2011), with the number of neighbors (k) set to the total number of haplotypes (2N for diploid samples). For missForest, we used the Python implementation (missingpy v0.2.0) of the nonparametric random forest-based imputation algorithm with default parameters (maximum of 10 iterations). Both methods were applied directly to the masked target samples without the use of any external reference panel. As above, imputation accuracy was evaluated by masking known genotypes and comparing the imputed results to the true values. These results were then compared to the accuracy achieved by our chimeric panel approach to assess relative performance in the absence of an external reference.

## Results
### Influence of chimeric reference panel size on imputation accuracy

We first examined how the size of the chimeric reference panel affects imputation accuracy across various proportions of data masking using the human dataset (Fig. 2a). As expected, higher masking proportions resulted in modest reductions in imputation accuracy. The relationship between panel size and accuracy varied with the masking level. For lower masking proportions (1%, 10%, and 20%), imputation accuracy remained relatively stable across a broad range of panel sizes, with 1% and 20% masking showing slight declines only after approximately 75 individuals, while 10% masking remained stable throughout the entire range tested. In contrast, at 30% masking, we observed a more pronounced pattern with performance peaking around 25–50 individuals before declining more noticeably. This suggests that the effect of panel size on imputation accuracy becomes more critical as the proportion of missing data increases. The differential responses across masking levels indicate that optimal panel size may depend on the expected level of missing data in the target dataset, with smaller panels (25–50 individuals, representing 12.5–25% of the total sample size) being sufficient when missing data is high (30%), while larger panels can be utilized without performance penalty when missing data is low to moderate (1–20%).

We then explored several parameters that could influence the compatibility between Retriever's chimeric reference panel and

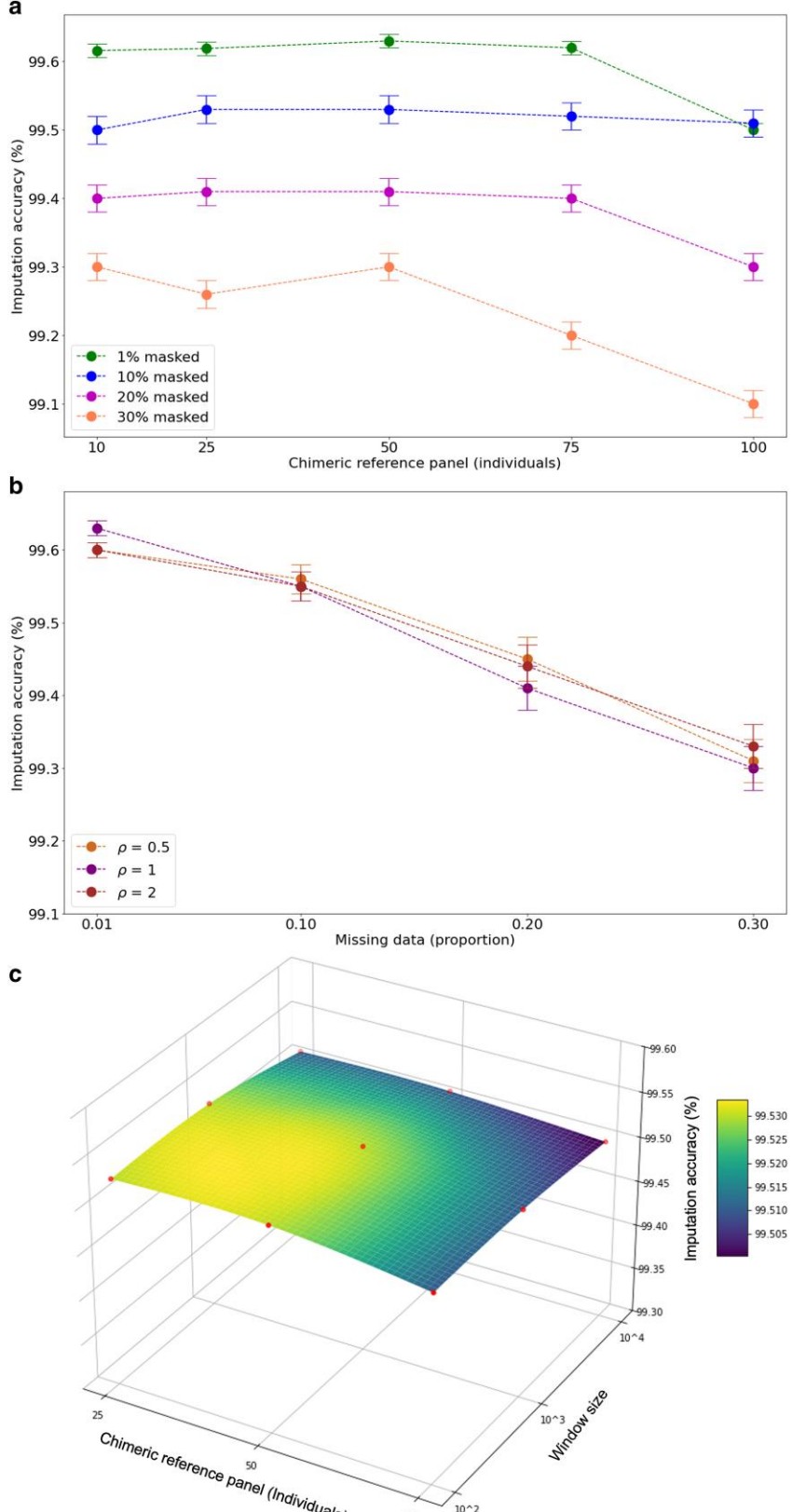

**Fig. 2.** a) Impact of Retriever's chimeric reference panel size on imputation accuracy. Imputation accuracy for *Homo sapiens* data obtained from the 1K Genomes Project, imputed in Beagle4. The chimeric reference panel size ranges from 10 to 100 individuals, selected from a target sample of 200 individuals. Each line represents a different proportion of missing data, ranging from 1% to 30% masked. b) Effect of recombination rate ($\rho$) on Beagle4 imputation accuracy when using a chimeric reference panel, evaluated at 30% missing data. Points represent the mean ± standard error from three independent replicates. c) Combined effect of chimeric reference panel size (25, 50, 75 individuals) and sliding window size on imputation accuracy, showing the trade-off between panel size and window length in maintaining linkage disequilibrium patterns. The gradient of the surface plot represents the optimal imputation accuracy achieved.

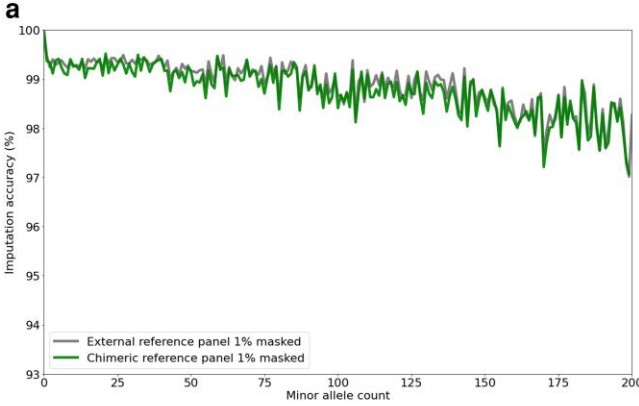

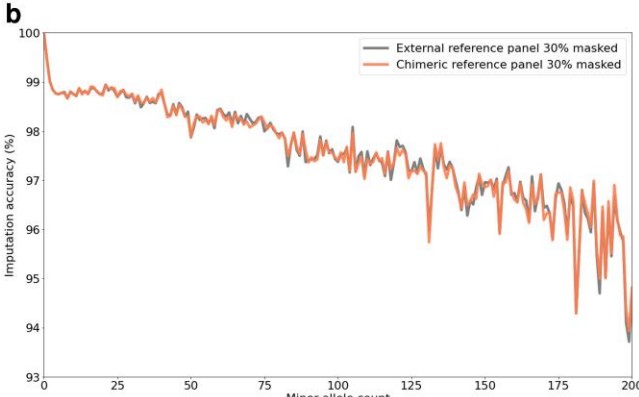

**Fig. 3.** Effect of minor allele count on imputation accuracy. *Homo sapiens* data obtained from the 1 K Genomes Project (2N = 400, $\pi$ = 0.027–0.034 Yasukochi et al. 2019) were masked at 1% (a) and 30% (b) and imputed with Beagle4 using either an external reference panel, or Retriever's chimeric reference panel, both with a panel size of 50 individuals. Imputed accuracies are binned according to the minor allele count from the original unmasked master file.

Beagle4's imputation algorithm. By default, Beagle4 assumes a uniform recombination rate of 1 centimorgan per megabase (cM/Mb). To assess its effect, we tested whether varying the recombination rate would influence imputation accuracy when using a chimeric reference panel. Across all levels of missing data, changes to the uniform recombination rate had a negligible impact on imputation accuracy (Fig. 2b), suggesting that Retriever's chimeric panels provide sufficient local haplotype information to maintain robust imputation performance even under suboptimal recombination rate assumptions. This insensitivity to recombination rate parameters represents an advantage for nonmodel organisms, where detailed genetic maps are often unavailable.

Next, we conducted a multivariate analysis to examine the effect of the sliding window size, which determines partition lengths, together with the number of chimeric individuals, on imputation accuracy. We tested nine combinations across three window sizes (0.1, 1, and 10 kb) and three chimeric panel sizes (25, 50, and 75 individuals) (Fig. 2c). Across these combinations, imputation performance remained relatively stable, demonstrating the robustness of our approach. The number of chimeric individuals was the most influential factor, accounting for 0.018% of the variance in imputation accuracy. Specifically, constructing 75 chimeric individuals led to a slight but consistent decrease in imputation accuracy across all window sizes. This suggests that larger chimeric panel sizes have a higher potential to create small partitions when sufficient complete genotype data becomes

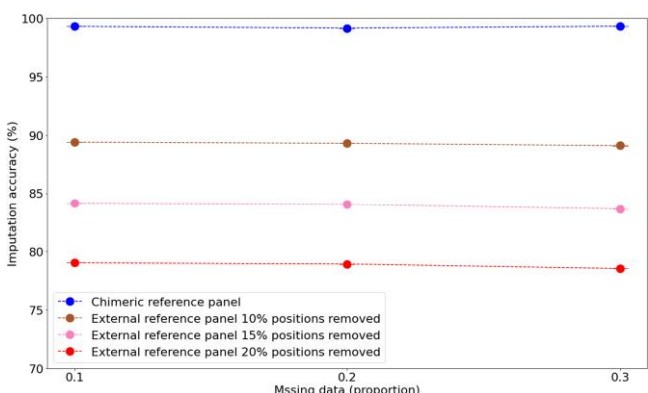

**Fig. 4.** Beagle4's performance on partially observed data demonstrating the cascading effects of incomplete reference panels. Target samples comprise 200 *Homo sapiens* individuals with 10% of their genotypes masked. Imputation of the target samples was compared using either a chimeric reference panel assembled with Retriever or an external reference panel with random positions removed across all samples (10%, 15%, and 20% positions removed). Points represent the mean ± standard error from three independent replicates.

limiting, potentially disrupting underlying linkage disequilibrium patterns and thus reducing imputation effectiveness. These results confirm that panel size optimization represents the primary consideration for users, with the optimal range of 25–50 individuals (12.5–25% of total sample size) providing the best balance between haplotype diversity and partition integrity.

## Imputation accuracy comparison with an external reference panel

We compared the imputation performance of Retriever's chimeric reference panel composed of 50 individuals against an external reference panel of equivalent size using the human dataset (Fig. 3a). The chimeric reference panel achieved imputation accuracies comparable to the external reference panel across all minor allele count bins, maintaining accuracy above 0.99 at low masking levels (1%) and 0.93 even at high masking levels (30%, Fig. 3b). This suggests that chimeric reference panels can effectively substitute for external panels of similar size, despite being constructed entirely from the target data itself. The equivalence in performance is particularly notable given that external reference panels typically require extensive construction resources and dedicated sequencing efforts to ensure complete genotype information across hundreds of individuals, while chimeric panels exploit the complementary distribution of missing data already present in the target dataset.

## Imputation of novel variants absent from external reference panels

A key limitation of conventional imputation approaches is their inability to impute genomic positions absent from the reference panel. Specifically, imputation software like Beagle discards any genomic positions present in the target samples but absent from the reference panel, effectively treating these novel variants as nonexistent rather than attempting to impute them. Thus, the presence of novel genotypes in target samples poses a significant challenge when external reference panels lack these variants. To quantify this effect, we systematically removed positions from the external reference panel while retaining them in the target samples (Fig. 4). This setup simulated incomplete reference coverage scenarios and assess the extent to which absent variants in the

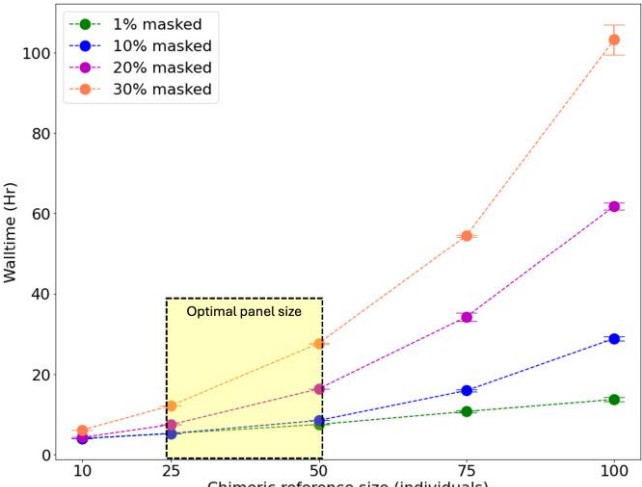

**Fig. 5.** Wall time plot showing the impact of varying proportions of masked data on chimeric reference assembly for human chromosome 1. Longer wall times indicate increased computational demand. The dotted box represents the optimal size of the chimeric reference panel, as determined from Fig. 2. Points represent the mean ± standard error from three independent replicates.

reference panel impair imputation accuracy. As expected, imputation accuracy using the external reference panel dramatically declined as the proportion of removed positions increased, with accuracy falling to below 80% when 20% of positions were absent from the reference panel. In contrast, Retriever's chimeric reference panel maintained close to perfect imputation accuracy. This fundamental advantage stems directly from Retriever's methodology. By constructing the reference panel from the target samples themselves, all variants present in the dataset—including those that would be novel relative to an external panel—are represented in the chimeric reference panel.

## Computational scaling and resource utilization

We evaluated the computational resources required for chimeric reference panel assembly under varying conditions of sample size and data masking (Fig. 5). Computational time (wall time) remained relatively stable when missing data (masked positions) was low (1%–10%), but increased substantially when missing data exceeded 20%. Importantly, for the optimal panel size range identified earlier (25–50 individuals from 200 samples), chimeric assembly required approximately 28 h, even with 30% missing data. These computation times represent a modest investment for most research groups. Although the wall-time presented reflects the chimeric assembly of a single chromosome, we recommend running all the chromosomes of a genome in parallel for improved time efficiency, as the computational requirements are relatively low.

Running on a standard high-performance computing environment with a single CPU core, the entire process can typically be completed overnight for most chromosomes. The computational time increases with the proportion of missing data in the dataset, as demonstrated in our scaling analysis. For context, these computational requirements are negligible compared to the resources needed to generate an external reference panel, which typically involves years of sequencing efforts, substantial financial investment, and collaborative networks—resources largely unavailable for nonmodel organism research. In general, Retriever should be accessible to research groups with limited resources, widening access to high-quality imputation software.

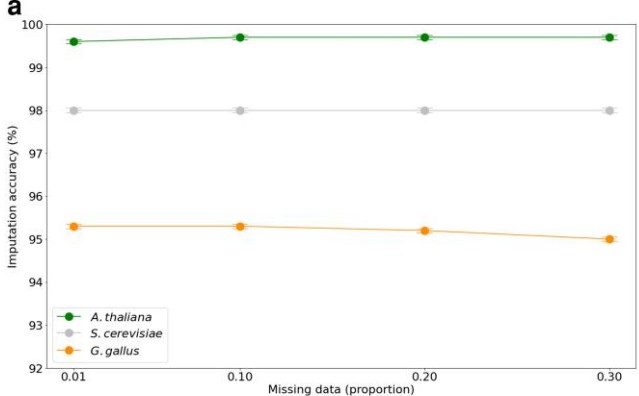

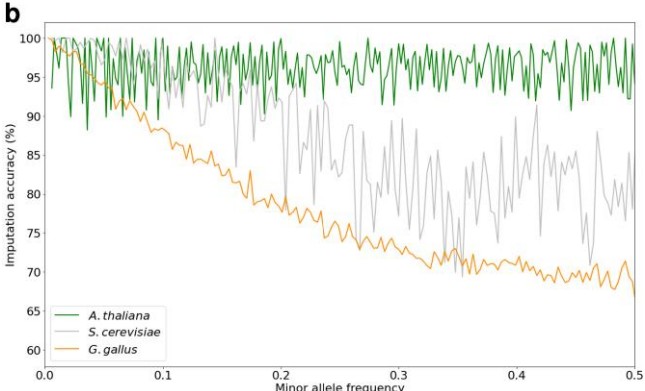

**Fig. 6.** a) Imputation accuracy of nonhuman model organisms. Chimeric reference panels, of 50 individuals, were constructed in Retriever with genomic data from *A. thaliana*, *G. gallus*, and *S. cerevisiae*, and imputed in Beagle4 across different proportions of masked (missing) data. Points represent the mean ± standard error from three independent replicates. b) Comparison of imputation accuracy across minor allele frequency bins for each species, based on datasets with 30% missing genotype data.

## Cross-species validation of imputation accuracy

To demonstrate Retriever's broad applicability, we extended our validation to diverse model organisms spanning plants (*A. thaliana*), animals (*G. gallus*), and fungi (*S. cerevisiae*) (Fig. 6a). Imputation accuracy consistently exceeded 95% across all species and levels of missing data, with minor variations likely corresponding to the underlying genetic diversity of each dataset. The *A. thaliana* dataset, derived from inbred lines with relatively homogeneous genetic structure, achieved the highest accuracy, whereas the *S. cerevisiae* dataset, representing 8 distinct populations, exhibited slightly lower but still excellent accuracy. The more genetically diverse *G. gallus* dataset, comprising 13 populations, showed the lowest accuracy, consistent with the expected negative relationship between population diversity and imputation performance. We also conducted an analysis of the diverse model organisms by stratifying imputation performance according to minor allele frequency (Fig. 6b). The results were consistent with expectations: greater population diversity within the samples likely introduces more complex population structures, which in turn increases the difficulty of accurately predicting genotype relationships.

We tested Retriever on *H. annuus* to evaluate its practicality in a nonmodel organism and its compatibility with downstream analyses, such as recombination rate inference (Fig. 7). The imputed dataset demonstrated a drastically lower estimated recombination rate compared to the original dataset with missing data,

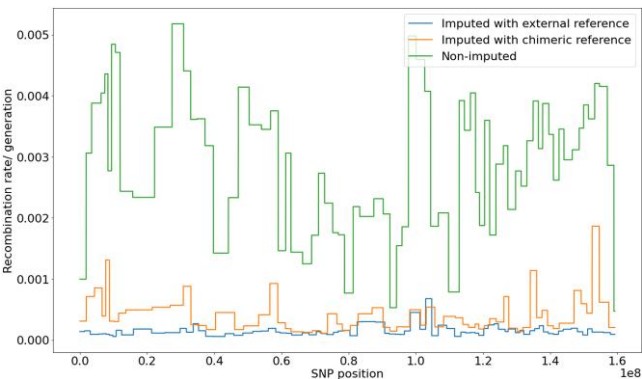

**Fig. 7.** Recombination rate inferred from *H. annuus* imputed and nonimputed data. Imputation is achieved using Retriever's chimeric reference panel (orange) and an external reference panel from a similar population (blue).

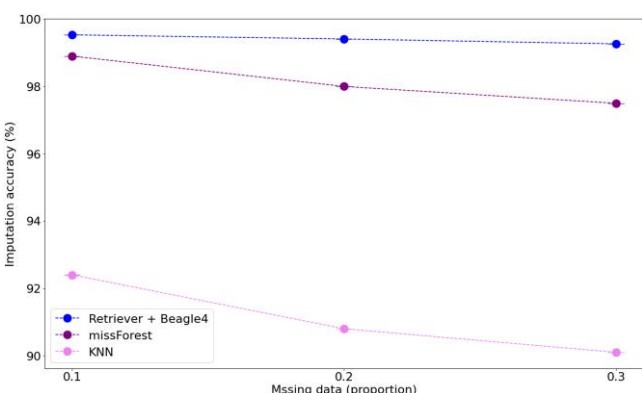

**Fig. 8.** Comparison of imputation accuracy between Retriever + Beagle4 and machine learning based imputation approaches (missForest and KNN) that are independent of an external reference panel. This analysis used the human dataset (*Homo sapiens*, 1000 Genomes Project) with varying proportions of masked genotypes. Points represent the mean ± standard error from three independent replicates.

consistent with results from imputation using an external reference panel obtained from the study by Todesco et al. (2020). This reduction in estimated recombination rates following imputation reflects the restoration of linkage information that was obscured by missing genotypes. Missing data can artificially inflate recombination rate estimates by breaking apparent linkage between variants that are actually in linkage disequilibrium, leading to overestimation of recombination events. By filling in missing genotypes based on local haplotype patterns, imputation restores the true underlying linkage structure, resulting in more accurate recombination rate estimates that better reflect biological. These findings highlight the importance of imputation for certain downstream population genetic and evolutionary analyses that are particularly sensitive to patterns of missing data.

## Comparison with reference-free imputation

Given that we developed Retriever as a framework to replace the requirement of an external reference panel, we also evaluated its performance by comparing the combination of Retriever and Beagle4 imputation against well-established reference-free imputation methods. We focused our comparison on machine learning-based approaches, including KNN and missForest, which are new alternatives to reference-based methods. While deep learning approaches may offer greater sophistication, their substantial computational demands and complex hyperparameter optimization requirements can present practical barriers for many research groups, particularly those working with nonmodel organisms and limited computational resources. Our chimeric reference panel approach consistently demonstrated higher overall imputation accuracy across varying levels of missing data, achieving accuracies that exceeded reference-free alternatives by 2–4% across most missing data scenarios (Fig. 8). Among the reference-free methods tested, missForest achieved relatively high imputation accuracy when the maximum number of iterations was set to ten, demonstrating the potential of iterative machine learning approaches. However, missForest's performance remained consistently below that of Retriever + Beagle4, likely reflecting the advantage of leveraging established population genetic principles through haplotype-based imputation rather than relying solely on pattern recognition from incomplete data. These results confirm that Retriever successfully bridges the gap between the accessibility of reference-free methods and the accuracy of reference-based approaches, providing an optimal solution for research on nonmodel organisms.

## Discussion

Genomic imputation faces a fundamental irony: the species that most need imputation—those with limited resources and sparse data—are precisely those excluded from the most accurate imputation methods. In this study, we developed Retriever, a novel framework that resolves this constraint by enabling the use of conventional imputation for species lacking external reference panels. Through rigorous benchmarking across plants, animals, and fungi, we demonstrated that Retriever's chimeric reference panel, combined with imputation in Beagle, maintains imputation accuracy above 95% across diverse taxonomic groups while eliminating the need for resource-intensive external reference panel construction. This advance democratizes access to sophisticated genomic imputation, extending these powerful analytical capabilities from a handful of well-funded model organisms to the full spectrum of biological diversity. Our results demonstrate that the complementary distribution of missing data across samples, combined with preservation of local linkage disequilibrium patterns, provides sufficient information to construct effective reference panels entirely from target datasets. Here, we examine the broader implications of these findings, explore the theoretical principles underlying Retriever's effectiveness, discuss its positioning within the landscape of genomic methodologies, and outline promising directions for future development.

## Methodological foundations and statistical principles of retriever

The fundamental innovation of Retriever lies in constructing a reference panel directly from the target data to be imputed. Our comparative analysis revealed that this chimeric reference panel performs comparably to external reference panels of equivalent size. This equivalence is remarkable, considering that external reference panels typically require extensive construction resources (Browning and Browning 2016; Sengupta et al. 2023), often involving dedicated sequencing efforts to ensure complete genotype information across hundreds of individuals. Both panel types exhibited similar performance patterns: imputation accuracy decreased as the minor allele count increased (reflecting greater genetic diversity) and as the proportion of missing genotypes increased (resulting in reduced available information about population structure).

When applying Retriever to diverse nonhuman organisms, we observed consistent imputation accuracy above 95% across species, with performance correlating with the complexity of population structure. *Arabidopsis thaliana*, derived from inbred lines, exhibited the highest and most stable accuracy across minor allele frequencies due to extensive linkage disequilibrium (LD) blocks created by self-fertilization and recent population bottlenecks. In contrast, *G. gallus* showed the steepest accuracy decline with increasing minor allele frequency, reflecting challenges from 13 distinct populations, shorter LD blocks, higher recombination rates, and haplotype diversity from domestication and breed admixture. *Saccharomyces cerevisiae* exhibited intermediate performance, consistent with its moderate subdivision across eight populations. These results demonstrate that imputation accuracy depends on the predictability of allelic associations: homogeneous populations with extensive LD facilitate accurate imputation, while diverse populations with complex demographic histories create more challenging prediction scenarios.

We recommend using Retriever primarily on samples composed of closely related individuals or within a single population, which aligns with common research practices of standard imputation software (Huang GH and Tseng 2014; Marino et al. 2022).

## Theoretical foundations of chimeric reference panel construction

The chimeric reference panel construction leverages the specific properties of genomic organization and the statistical distribution of missing data. Retriever exploits the nonrandom patterns of linkage disequilibrium, haplotype block structure (Dekeyser et al. 2023; Mochurad and Horun 2023), and recombination rate variation (Smukowski and Noor 2011) across the genome, alongside the probabilistic complementarity of sequencing coverage across multiple samples (Nunney 2001). By "probabilistic complementarity of sequencing coverage," we refer to the statistical phenomenon whereby the patterns of missing data tend to vary stochastically across samples in next-generation sequencing datasets.

As sequencing depth varies across genomic regions due to factors such as GC content (Aird et al. 2011; Benjamini and Speed 2012; Huang H and Knowles 2016), sequence complexity (Sims et al. 2014), structural features (Muyas et al. 2019), and stochastic sampling, missing genomic data rarely follows an entirely random pattern (Wong et al. 2019). This nonrandom distribution creates a favorable scenario: the probability that any specific position will be missing in all samples decreases exponentially as the sample size increases. Consequently, despite substantial missing data in individual samples, the collective dataset retains information at nearly all genomic positions across different subsets of individuals. Retriever exploits this fundamental statistical property to construct complete chimeric reference panels. Our results demonstrate robust performance with up to 30% missing data, suggesting that traditional preprocessing thresholds (often set at 20% missingness) may be unnecessarily conservative when using Retriever (see Methods). We recommend removing only those positions where missingness is so extreme that no subset of individuals has complete data, as these positions would prevent the construction of a chimeric panel entirely. This represents a departure from conventional quality control practices, potentially retaining thousands of additional informative sites that would traditionally be discarded. For users concerned about computational time, filtering positions with more than 30% missingness can improve efficiency without sacrificing accuracy, although our results indicate that this is optional rather than required.

Researchers should remain cognizant of potential biases when interpreting results from regions known to be problematic for sequencing technologies.

The chimeric reference panel construction leverages the natural organization of genetic variants in genomes. Genetic variants are not randomly distributed (Cheung et al. 2011; Dong et al. 2021), they show patterns of association with nearby variants that reflect the population's evolutionary history, including past recombination events and selection pressures (Edwards and Beerli 2000; Hellenthal and Stephens 2006). Retriever's sliding window approach seeks to preserve these important genetic patterns by identifying genomic partitions with complete genotype information. However, this preservation is contingent upon the distribution of missing data, which could disrupt these associations. If missing genotypes occur nonrandomly and cluster in regions of strong linkage disequilibrium (LD) or at functional elements, the effectiveness of the sliding window approach may be compromised. Our empirical validation suggests that under typical sequencing scenarios, sufficient complementarity exists to maintain the integrity of the local LD structure.

Our optimization testing revealed that chimeric panels with 25–50 individuals (representing 12.5–25% of our 200 test samples) achieved the highest accuracy. This finding is logical when considering the trade-offs involved. Larger panels can capture more genetic diversity but force the algorithm to use smaller window sizes to find enough complete data segments. When windows become too small, they break up the natural patterns of genetic association that help predict missing genotypes. Based on these findings, we recommend setting the chimeric reference panel size to 25–50 individuals, or approximately 12.5–25% of the total sample size for datasets significantly larger or smaller than our test set.

Retriever also takes advantage of the natural constraints on genetic variation. The combinations of genetic variants we observe are not random—they are shaped by evolution, natural selection, and population history (Akey et al. 2004; Smukowski and Noor 2011; Zhang et al. 2013). These constraints limit the possible genotypes when predicting missing data, making imputation more tractable. While rare genetic variants are generally harder to impute correctly (Lau et al. 2024) (they provide less information for the algorithm to work with), our chimeric reference panel method performs similarly to traditional reference panels across all variant frequencies, suggesting our approach successfully captures the underlying genetic patterns that determine which variant combinations are biologically likely.

## Implications of stochastic sampling in chimeric panel construction

The assembly of the chimeric reference panel involves randomly selecting individuals from each "bucket" of complete genotypes within defined genomic windows. This stochastic component introduces both advantages and potential limitations. On the one hand, the randomization process prevents ascertainment biases while maintaining the fundamental property of complete genotype information within each window. However, random selection could remove rare variants and potentially disrupt trans-window haplotype continuity when transitions occur between adjacent genomic regions. Our empirical analyses suggest that this concern has minimal practical impact on the accuracy of imputation. This resilience likely reflects the primacy of local LD patterns in determining imputation accuracy; that is, preserving complete haplotype information within windows appears more critical than maintaining long-range haplotype continuity across window boundaries. Nevertheless, future iterations of

Retriever could implement more sophisticated sampling strategies that consider haplotype similarity at window boundaries. For instance, we could prioritize matching individuals across adjacent windows when possible. If we have sampled individuals (3, 7, 1, 2) in one window and individuals (8, 11, 5, 7) in the next, we could create concatenated chimeric reference sequences by connecting individuals with shared identities first (like individual 7), then completing the remaining connections (e.g. 3–8, 7–7, 1–11, and 2–5). This approach would potentially enhance trans-window haplotype continuity while maintaining the core advantages of the chimeric assembly process.

### *Advantage of capturing novel variants*

A significant advantage of Retriever over conventional methods is its ability to impute novel variants absent from external reference panels. As sequencing technologies advance, newer studies often identify variants or indels not present in existing reference resources (Satam et al. 2023). Conventional imputation approaches that rely upon external reference panels discard these novel positions, potentially losing valuable information (Lau et al. 2024). By contrast, Retriever retains and processes all variants detected in the target dataset during the imputation process. This means population-specific variants and newly discovered polymorphisms are incorporated into the imputation framework rather than being systematically excluded. This inclusion is particularly valuable for capturing functionally relevant variation that might be population-specific or absent from established reference resources. Our analysis demonstrates that Retriever maintains significantly higher imputation accuracy compared to external reference panels when target samples contain novel genomic positions, ensuring that valuable information is preserved rather than discarded.

## Resource utilization and efficiency

The computational demands of assembling a chimeric reference panel are modest compared to the resources required for constructing an external reference panel (1000 Genomes Project Consortium et al. 2015; Tadaka et al. 2021). Even with 30% masked data, assembling an optimal chimeric panel typically takes less than 30 h of wall time. Although this represents a significant computational runtime, it is a negligible cost compared to the years of sequencing efforts, substantial financial investment, and collaborative networks required to build an external reference panel—resources typically unavailable for nonmodel organism research.

By shifting the resource burden from wet-lab sequencing to computational analysis, Retriever provides an economical solution for genomic imputation while maintaining high accuracy. This accessibility has particular significance for conservation genomics (Fuentes-Pardo and Ruzzante 2017), biodiversity research (Escalante et al. 2014), and studies of neglected crop species (Liu 2011; Esposito et al. 2016; Ashraf et al. 2022), where resource limitations have historically constrained methodological options.

## Comparative advantages of retriever over other methodologies

Contemporary genomic imputation methodologies can be categorized along a methodological spectrum, ranging from reference-based approaches to reference-free strategies, each with distinct conceptual foundations and operational constraints. Retriever occupies a strategic intermediate position, synthesizing advantageous elements from both paradigms while addressing their respective limitations.

### *Reference-based methodologies: statistical power with resource limitations*

Reference-based approaches, exemplified by established frameworks such as Beagle (Browning and Browning 2016), IMPUTE2 (Scheet and Stephens 2006), IMPUTE5 (Rubinacci et al. 2020), Minimac (Howie B et al. 2012), and more recently EagleImp (Wienbrandt and Ellinghaus 2022) and Minimac4 (Das et al. 2016), rely on comprehensive external reference panels and sophisticated statistical models. These methods predominantly implement hidden Markov models (HMMs) or positional Burrows-Wheeler transforms to infer unobserved genotypes through probabilistic modeling of haplotype structure. Their statistical power derives from capturing population-level patterns of linkage disequilibrium encoded within reference panels (Phocas 2022a), which comprise hundreds to thousands of fully genotyped individuals.

However, reference-based approaches face substantial constraints. EagleImp and IMPUTE5, despite offering state-of-the-art accuracy for human genomics, require not only reference panels but also genetic maps and phased reference panels, creating resource requirements that restrict their applicability beyond model organisms. Even Beagle4, which exhibits greater flexibility, shows diminishing returns when the reference panel size decreases below certain thresholds—a particular challenge for non-model organisms, where comprehensive genomic resources remain unavailable. Moreover, these methods discard genomic positions absent from reference panels, excluding potentially significant novel variation. This limitation creates a methodological blind spot for population-specific variants, which often hold particular functional or evolutionary significance (Huang GH and Tseng 2014). This constraint becomes increasingly problematic as sequencing technologies advance, revealing previously undetected genetic diversity in a progressive manner.

### *Reference-free approaches: accessibility with black-box complexity*

At the opposite end of the methodological spectrum, reference-free approaches have emerged to address these accessibility limitations. Recent implementations include machine learning frameworks such as GRUD (Chi Duong et al. 2023), which employs gated recurrent units with adversarial training; Two-Stage DAE +RNN (Kojima et al. 2024), which combines denoising autoencoders with recurrent neural networks; and SOM-based imputation (Mora-Márquez et al. 2025), which implements self-organizing maps for pattern recognition. Several studies have achieved SNP imputation through machine learning approaches such as the use of KNN (Schwender 2012) and missForest (Stekhoven and Bühlmann 2012) with performance ranging from 93% to 97%.

Our results were consistent with findings from previous studies; however, the combination of the Retriever's chimeric reference panel with Beagle4 imputation consistently achieved the highest accuracy. Notably, missForest has also demonstrated strong potential, suggesting that iterative imputation approaches are promising. One possible limitation of missForest lies in the initial step of temporarily filling missing values with the mode genotype, which can introduce a small amount of error. This initial error may propagate through subsequent iterations, potentially limiting the maximum achievable accuracy. This limitation could potentially be addressed by integrating the chimeric reference panel's ability to capture the population structure, combined with an iterative imputation framework. Further studies are needed to explore this combined approach and its impact on imputation performance.

**Table 1.** Comparison of retriever and Beagle4 with reference-based and reference-free imputation approaches.

| | Retriever + Beagle4 | Reference-based | Reference-free ML |
|---|---|---|---|
| Core approach | Assembles a chimeric reference panel from target samples | Uses external reference panels comprising fully genotyped individuals | Learns patterns directly from incomplete data without reference panels |
| Accuracy | >95% across diverse taxa and masking levels | >98% for model organisms; performance degrades for nonmodel organisms | 93%–97% depending on implementation and data characteristics |
| Novel variant handling | Preserves and imputes all genomic positions in target samples | Cannot impute positions absent from reference panel; novel variants discarded | Variable capability; some may impute genotypes in novel regions |
| Resource requirements | Moderate computational resources; no external panel needed | External panel requiring extensive sequencing efforts across hundreds of individuals | Substantial computational resources for model training; no external panel needed |
| Biological interpretability | Preserves explicit haplotype structures and linkage disequilibrium patterns | Direct modeling of population genetic parameters | Abstract learned representations without direct population genetic interpretation |
| Primary application | Nonmodel organisms lacking reference resources | Model organisms with established external reference panels | Nonmodel organisms lacking reference resources |
| Implementation complexity | Moderate; requires basic configuration of window parameters | High; dependent on reference panel availability and quality | High; requires expertise in machine learning methodologies and hyperparameter optimization |

These methods circumvent reference panel requirements by learning genotype patterns directly from incomplete data. Their key advantage lies in accessibility, eliminating the infrastructure demands of reference panel construction. However, they introduce distinct limitations: complex hyperparameter optimization requirements, substantial computational resources for model training (often exceeding 50–100 h for genome-wide applications), and limited biological interpretability of their underlying statistical representations (Naito and Okada 2024). Furthermore, these methods typically demonstrate 2–5% lower accuracy compared to reference-based approaches when evaluated on datasets with available reference panels (Kojima et al. 2024). This performance gap widens particularly for rare variants and regions with complex haplotype structures, where deep learning models struggle to capture subtle biological patterns from limited examples (Jiang et al. 2021).

### Retriever's chimeric framework: combining methodological strengths

Retriever strategically bridges reference-based and free imputation paradigms through its chimeric reference panel construction. Unlike both methodological categories, Retriever uniquely maintains the established statistical frameworks of reference-based methods while eliminating the need for external references. Further, unlike the abstract learned representations of machine learning approaches, Retriever preserves an explicit representation of haplotype structures. Retriever also imputes all genomic positions observed in the target dataset, including novel variants systematically excluded by reference-based methods, and requires minimal configuration compared to the extensive hyperparameter tuning demanded by machine learning approaches. Finally, it shows consistently high accuracy (>95%) across diverse taxa, approaching the performance of optimal reference-based methods while exceeding typical reference-free implementations by 1–3% (Chi Duong et al. 2023; Mora-Márquez et al. 2025)

We also demonstrated the use of Retriever in a nonmodel organism, *H. annuus*, where the imputed data showed a drastic reduction in estimated recombination rates. While this outcome may not generalize to all scenarios, it serves as an example of how imputation can be performed in nonmodel species and can help mitigate artificial inflation or deflation in downstream analyses. *Helianthus annuus* was selected for this demonstration because the dataset contains a high proportion of missing genotypes, making it unsuitable for further masking. Moreover, unlike model organisms, *H. annuus* lacks access to a large and well-curated reference panel, making it an ideal candidate to showcase Retriever's utility in such contexts. Although we included a comparison to data imputed using an external reference panel, we do not consider this to represent the ground truth. The external panel used in the study requires more detailed information, such as its composition and quality. Therefore, while it provides a useful point of comparison, it should not be interpreted as a definitive benchmark.

Retriever's chimeric methodology offers distinct advantages in various contexts. For integrative studies involving both model and nonmodel organisms, Retriever enables methodological consistency across datasets with varying resource availability. In conservation genomics, where sample availability necessarily limits the construction of reference panels, Retriever can extract information from sparse data without sacrificing the necessary rigor in downstream statistical analysis. For novel variant analysis in clinical genomics, Retriever's preservation of observed genomic positions potentially captures functionally relevant variation that can be excluded by conventional approaches. The comparable performance between chimeric and external reference panels of equivalent size (Table 1) demonstrates that our approach successfully preserves essential population genetic information required for accurate imputation while substantially reducing barriers to implementation. This combination of accessibility, interpretability, and performance positions Retriever as a valuable bridge, extending genomic analysis capabilities beyond traditional model systems. However, Retriever was designed not to replace existing imputation methods using large external reference panels but as an alternative for studies that lack the external panel to perform imputation, particularly in nonmodel organisms.

Our method assumes that the samples are derived from a single, genetically similar population (Blomberg and Todorov 2025). However, this assumption may not always hold in real-world applications, particularly when the ancestry or taxonomy of the samples is unknown (Chikhi et al. 2010). Therefore, users should carefully assess the suitability of their dataset for imputation and determine whether the samples can reasonably be

considered as originating from a single population. Additionally, the method assumes that the input data are of high quality, which must be ensured by the user during the data preprocessing and quality control phase. These considerations are not unique to our method. Still, they are critical when selecting any imputation approach, as the accuracy and reliability of imputation are strongly influenced by the underlying data quality and population characteristics.

### Future directions

While Retriever currently demonstrates strong performance across diverse taxa, several promising directions can enhance its methodology. Currently, Retriever assembles the chimeric reference panel using a random selection of complete genotype partitions, but future versions could implement a more sophisticated partition ranking system. By prioritizing genomic partitions based on their likelihood of containing rare alleles or evolutionarily significant variants, Retriever could enhance imputation of low-frequency genetic variation while increasing overall panel information content. Integration with machine learning approaches represents another promising direction. Incorporating neural network architectures such as attention-based models or graph neural networks to learn complex patterns from the chimeric reference panel could potentially improve imputation in regions with limited linkage disequilibrium or complex structural variation. This hybrid approach would combine the biological interpretability of reference-based methods with the pattern recognition capabilities of machine learning.

These methodological enhancements would expand Retriever's applicability to increasingly complex genomic contexts and data types. Retriever also has potential applications in several challenging genomic contexts. For polyploid species (e.g. wheat, potato, cotton), which represent many agriculturally important crops, adapting the sliding window approach to accommodate higher ploidy levels could significantly enhance genomic research in these systems. For ancient DNA studies, where highly fragmented and damaged genetic material creates extreme missing data patterns, Retriever's approach could be combined with damage-aware models to improve the recovery of population genetic information from archaeological samples. In clinical genomics, where rare variants often have outsized functional importance, Retriever could be extended to incorporate functional annotation data, prioritizing imputation accuracy in regions with potential phenotypic consequences. For conservation genomics, adaptation to handle extremely small sample sizes through integration with phylogenetic information could enhance genetic management of endangered species with limited sampling opportunities.

As long-read sequencing technologies become increasingly prevalent, extending Retriever to leverage their unique error profiles and capacity for direct haplotype phasing represents another critical direction for future development. While the current algorithm is optimized for short-read data characteristics, modifications to the window selection approach could take advantage of the extended contiguity information provided by long reads, potentially enabling larger window sizes and more precise boundary definition. The complementary strengths of short-read coverage and long-read contiguity could be combined to further enhance imputation accuracy, particularly for complex structural variants that remain challenging for current approaches. Additionally, while our current implementation maintains fixed panel sizes for compatibility with existing imputation software, future versions could explore dynamic sizing strategies that leverage all available complete data in regions of high coverage, potentially improving accuracy for rare variants while maintaining computational efficiency.

## Conclusion

Retriever represents a significant advancement in genomic imputation methodology, enabling high-accuracy genotype imputation for nonmodel organisms without the need for external reference panels. By constructing chimeric reference panels directly from target samples, our approach maintains imputation accuracy exceeding 95% across diverse taxa while preserving essential patterns of linkage disequilibrium. The method demonstrates particular strength in retaining novel variants absent from traditional reference resources. The effectiveness of Retriever rests on the complementary distribution of missing data across samples, the nonrandom structure of linkage disequilibrium in genomes, and the constrained patterns of allelic variation imposed by population genetic processes. Together, these principles enable the construction of effective reference panels from partially observed data, circumventing the resource-intensive requirements of conventional approaches. Retriever transforms a historical limitation of genomic research into an opportunity, making sophisticated imputation accessible beyond model organism research. In doing so, it contributes to a more inclusive genomic science capable of addressing biological questions across the full diversity of life.

## Data availability

The source code for Retriever is publicly available on GitHub (https://github.com/uqmzhou8/Retriever.git) and is released under the MIT license. The repository includes comprehensive documentation, installation instructions, and example scripts to facilitate implementation. All genomic datasets used for validation in this study were obtained from publicly available repositories. Human genomic data were sourced from the 1000 Genomes Project Phase 3 release (http://www.internationalgenome.org/). *Arabidopsis thaliana* sequence data were obtained from the 1001 Genomes Consortium hosted by the Max Planck Institute for Developmental Biology (https://1001genomes.org/). *G. gallus* genome sequences were accessed through the study by Cho et al. (2022) available at the European Nucleotide Archive (Project name: PRJEB44919 2021-05-15). *Saccharomyces cerevisiae* genomic data were obtained from Sardi et al. (2018) with accession number PRJEB24747 in EBI. *Helianthus annuus* genomic data were obtained from Todesco et al. (2020) available at https://sunflowergenome.org/.

The specific parameter settings, analytical procedures, and simulated masking patterns used in our evaluation of Retriever are described in detail within the Methods section. Supplementary scripts used for benchmarking, including VCF file processing, performance evaluation, and statistical analysis, are available in the GitHub repository under https://github.com/uqmzhou8/Retriever.git.

Researchers implementing Retriever or reproducing our analyses are requested to cite this paper and the relevant data sources according to the respective citation guidelines of their field. For additional inquiries regarding specific implementations or advanced configurations, please contact the corresponding author.

Supplemental material available at GENETICS online.

## Acknowledgments

We thank the editor, Yun Li, and the two reviewers for their constructive comments, which significantly improved the clarity of

the manuscript. We are grateful to members of the Ortiz-Barrientos Laboratory for their feedback on the figures and to Andrew D. Letten for his input during the validation process.

## Funding

This work was supported by an Australian Research Council grant awarded to D.O-B (FT200100169) and the Australian Research Council Centre of Excellence for Plant Success in Nature and Agriculture (CE200100015).

## Conflicts of interest

None declared.

## Author contributions

DOB conceptualized the chimeric reference panel approach for genomic imputation, developing the hypothesis that the statistical complementarity of missing data patterns across samples exists. MZ implemented this concept by developing the Retriever algorithm, conducting simulations, performing analyses, and creating visualizations to support the concept. MZ received methodological guidance on statistical validation, evolutionary implications, and population genetics from the research team. MJ, JE and DOB supervised the project and contributed to experimental design and results interpretation. MZ wrote the initial manuscript draft. All authors contributed to the manuscript revision, enhancing the theoretical framework, methodology, and discussion to produce the final version.

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

*Editor: Y. Li*