## [Peer Review File · Genetics]

Chimeric Reference Panels for Genomic Imputation

Meikun Zhou, Maddie James, Jan Engelstaedter, and Daniel Ortiz-Barrientos

NOTE: The reviews and decision letters are unedited and appear as submitted by the reviewers.

In extremely rare instances and as determined by a Senior Editor or the EIC, portions of a review may be redacted. If a review is signed, the reviewer has agreed to no longer remain anonymous.

The review history appears in chronological order.

Review Timeline:

Submission Date:	2025-04-14
Editorial Decision:	2025-06-02
Resubmission Received:	2025-07-21
Editorial Decision:	2025-08-20
Revision Received:	2025-09-14
Accepted:	2025-09-17

June 1, 2025

GENETICS-2025-308076
Chimeric Reference Panels for Genomic Imputation

Dear Dr. Zhou:

Two experts in the field have reviewed your manuscript, and I have read it as well. The reference-free imputation framework proposed has multiple merits that are potentially advantageous over or complementary to existing methods. While your manuscript is not currently acceptable for publication in GENETICS, we would welcome a substantially revised manuscript. Both reviewers have comments and concerns to be addressed in a revised manuscript. You can read their reviews at the end of this email. We look forward to receiving your revised manuscript. Please let the editorial office know approximately how long you expect to need for revisions.

Upon resubmission, please include:

1. A clean version of your manuscript;
2. A marked version of your manuscript in which you highlight significant revisions carried out in response to the major points raised by the editor/reviewers (track changes is acceptable if preferred);
3. A detailed response to the editor's/reviewers' feedback and to the concerns listed above. Please reference line numbers in this response to aid the editor and reviewers.

Your paper will likely be sent back out for review.

Additionally, please ensure that your resubmission is formatted for GENETICS
<https://academic.oup.com/genetics/pages/general-instructions>

Follow this link to submit the revised manuscript: Link Not Available

Sincerely,

Yun Li
Associate Editor
GENETICS

Approved by:
Konrad Lohse
Senior Editor
GENETICS

Reviewer #1 :

Summary:

The authors present Retriever, a reference-free genotype imputation framework designed for non-model organisms, which leverages a sliding-window approach to construct a chimeric reference panel from the target data. The authors compared Retriever's performance with external reference panels using Beagle4 and performed analysis on multiple taxa. The method addresses the practical limitation that external reference panels are often unavailable for non-model species.

Major Comments:

1. The method assumes that while individual samples may have missing genotypes, the collective dataset contains complete data across the genome. While this may hold for datasets with randomly distributed missingness, I question how often this assumption holds in real-world non-model datasets, which may have systematic biases (e.g., due to low coverage, population structure, or technical artifacts). The authors should clarify to what extent their assumption holds empirically across the species tested, and whether it generalizes beyond the datasets evaluated.
2. The authors state that their approach is aimed at non-model organisms. However, the manuscript would benefit from more explicit contrast with conventional imputation settings—such as human datasets where genotype arrays are used, and external panels are often available. It would be helpful to emphasize that proposed method is not meant to replace those methods but to extend imputation capabilities where there is no external reference available.
3. The approach assumes genotype calls are error-free (as stated in the Methods), target samples are genetically similar (i.e.,

within populations), and high-quality pre-processing has been performed. These assumptions are reasonable but should be more prominently discussed as potential limitations or caveats to consider.

4. Does the chimeric panel need to be of fixed size across the genome? The algorithm seems to randomly subsample when there are more samples than required. Could it instead use all available data when possible, to improve accuracy or robustness? Would dynamic sizing help?

5. In the comparison between Retriever and external panel-based imputation, the authors remove certain variants from the external reference panel while retaining them in the target samples. This setup inherently favors Retriever and may not reflect realistic use cases. A more informative comparison would involve scenarios grounded in actual differences between sequencing platforms or study designs. The authors should clarify the rationale behind this simulation strategy and discuss its limitations. In particular, it is unclear why, in a real-world setting, a researcher would choose an external reference panel that has lower genomic coverage than the target dataset.

6. The authors report that chromosome 1 takes ~28 hours for chimeric panel construction under high missingness. This raises concerns about scalability to genome-wide analyses. Figure 5 would benefit from more annotation to clarify whether the times reflect single chromosome runs, and whether parallelization or optimization strategies were considered.

7. In Figure 2, the differences in imputation accuracy across chimeric panel sizes are within ~0.5%, which is small. Are these differences statistically or practically significant? Some additional interpretation would help.

8. The sliding window size appears to default to 1kb. How sensitive is imputation accuracy to this choice? Have the authors tested different window sizes or overlap strategies?

9. The study compares Retriever to external reference panel-based imputation using Beagle4, but lacks comparison with other reference-free methods that were discussed. Some benchmark will be very helpful.

10. The core innovation lies in assembling a chimeric reference panel from complete partitions in the target data. While this is a useful contribution, the imputation itself still relies entirely on Beagle4. The manuscript should better highlight the specific methodological contribution (i.e., reference construction) and clarify that Retriever is a preprocessing framework, not a novel imputation algorithm per se.

Reviewer #2 :

Summary:

This article proposed Retriever, an approach that builds a "chimeric" reference panel for downstream genomic data imputations by connecting non-overlapping genetic partitions, or windows, where only individuals with non-missing genotype data are selected. As advantages, this approach not only obviates the requirement of building an external reference panel that needs a large sample size and cost, but also shows unique strength in imputation of novel variants that are absent from previous external reference panels. Furthermore, this approach is especially useful for non-model organisms that lack large-scale genomic data. When compared with reference-free ML approaches that also don't require an external reference panel, it is claimed that Retriever is more biologically interpretable and less complex in implementation.

However, this innovative approach still has limitations, and several comments are listed as follows:

Major Comments:

1. In the article, the author compares the proposed approach with both reference-based and reference-free approaches, as shown in Table 1. However, the imputation accuracy (> 95%, >98%, 93-97%) is not fully computed and shown in the result section. During the benchmark, the author has only compared the imputation result between the chimeric and external reference panels. Although the author mentioned that "reference-free approaches demonstrate typically 2-5% lower accuracy compared to reference-based approaches" based on previous studies, no actual numerical results are provided in the article to compare Retriever with any reference-free approaches. Therefore, a more comprehensive benchmark of the Retriever with both reference-based and -free methods using the same training data and test data may be needed to strengthen the current message in Table 1.

2. The paper's imputation results are mostly obtained through the integrated use of Retriever and Beagle4.1 without a genetic map file. From their documents (https://faculty.washington.edu/browning/beagle/beagle_4.1_21Jan17.pdf), it seems that a constant recombination rate of 1 cM / Mb will be set if no genetic map files are specified. Further clarification/comparison might be needed on how much this will affect the imputation performance when using the Retriever.

3. Throughout the article, the authors explored and listed several factors (apart from the missing data proportion) that will impact the imputation performance when using Retriever: (1) window size, (2) number of individuals (sample size), and (3) minor allele count (genetic diversity) of the selected population or organism. However, the effect of each factor is measured independently. A potential joint analysis will help users better understand the interaction between these factors and select the optimal choices

based on their target data.

4. In Materials and methods, the authors noted that preprocessing the input VCF files requires removing multiallelic sites, which is quite common in genomic-based tools. But from the algorithmic perspective (Figure 1), the Retriever seems to be able to handle multiallelic sites. Could Retriever do that with the current implementation? Also, could Retriever handle organisms with polyploidy? It would be great to see the idea from the Retriever used on a even broader scale.

5. In cross-species validation, the author suggests that organisms with higher genetic diversity will result in poorer imputation performance. To represent genetic diversity in the three non-human organisms (*Arabidopsis thaliana*, *Gallus gallus*, and *Saccharomyces cerevisiae*), having a measurable metric like minor allele count would be better, which will help users estimate the imputation accuracy for the data from their organisms.

6. It's great that the GitHub README gives instructions on installing external libraries for the Retriever. Still, a step-by-step tutorial with shell/Python code from preprocessing to output would be more straightforward for users. If possible, adding a tiny data example would be more helpful.

Minor Comments:

1. The line after the title "Retriever's chimeric framework: Combining methodological strengths" got a typo. Instead of "Retriever strategically bridges reference- and -free and imputation paradigms ...", it should be "Retriever strategically bridges reference-based and -free imputation paradigms ..."

2. In the Algorithm 1, line 8, the "(-3)" between "missing data" and "occurs" seems confusing. Please correct it if it is a typo.

3. The figures are not referenced anywhere in the text.

Associate Editor Comments:

Response to Reviewers

Dear Editor, and Reviewers,

Thank you for taking the time to review our manuscript. We have carefully addressed all comments given by both reviewers and have referenced the corresponding changes in the track changes document. We would like to express our gratitude to the reviewers for their constructive suggestions, which have significantly improved the quality of the manuscript. In response, we conducted additional analyses to demonstrate how Retriever can be effectively used as a preprocessing step prior to imputation, ultimately enhancing downstream analyses. Furthermore, we have included a case study highlighting the benefit of performing imputation in non-model organisms using Retriever, compared to analyses conducted without imputation, shown in Fig. 7 and page 11 lines 11-28. All page and line references are referring to the track changes document, which have most of the figures at the end of the document but this is not the case for both the old and revised manuscripts.

We hope the revisions meet the expectations of the reviewers and editorial board, and we look forward to your feedback.

Regards, Meikun Zhou (on behalf of all authors)

Editor's Comments

Dear Dr. Zhou:

Two experts in the field have reviewed your manuscript, and I have read it as well. The reference-free imputation framework proposed has multiple merits that are potentially advantageous over or complementary to existing methods. While your manuscript is not currently acceptable for publication in GENETICS, we would welcome a substantially revised manuscript. Both reviewers have comments and concerns to be addressed in a revised manuscript. You can read their reviews at the end of this email. We look forward to receiving your revised manuscript. Please let the editorial office know approximately how long you expect to need for revisions.

Reviewer #1

Summary: The authors present **Retriever**, a reference-free genotype imputation framework designed for non-model organisms, which leverages a sliding window approach to construct a chimeric reference panel from the target data. The authors compared **Retriever**'s performance with external reference panels using **Beagle4** and performed analysis on multiple taxa. The method addresses the practical limitation that external reference panels are often unavailable for non-model species.

Major Comments:

1. The method assumes that while individual samples may have missing genotypes, the collective dataset contains complete data across the genome. While this may hold for datasets with randomly distributed missingness, I question how often this assumption holds in real-world non-model datasets, which may have systematic biases (e.g., due to low coverage, population structure, or technical artifacts). The authors should clarify to what extent their assumption holds empirically across the species tested, and whether it generalizes beyond the datasets evaluated.

Response: Thank you for this important question about missing data assumptions. We recognize that our initial framing using MAR/MCAR terminology was imprecise, as these frameworks typically describe patterns of missingness within target datasets themselves, whereas our study primarily addresses reference panel completeness—whether external panels contain all variants present in target samples. Following internal discussion, we decided to move away from technical MAR/MCAR terminology to make the manuscript more accessible to general readers. Instead, we provide a clear plain-English description of the two scenarios **Retriever** addresses: (1) complete reference coverage where external panels contain all variants, and (2) incomplete reference coverage where external panels lack variants present in target samples. This is explained in the 'Imputation of novel variants absent from external reference panels' section (page 6 lines 58-69) and demonstrated empirically in Figure 4.

This situation frequently arises in real-world applications when target samples are sequenced with newer technologies that reveal variants missed during original reference panel construction, or when studying populations not well-represented in existing reference resources. **Retriever** addresses this fundamental limitation by constructing reference panels

directly from the target data, ensuring all observed variants are retained and processed during imputation rather than being systematically excluded.

Our preprocessing workflow (page 2, lines 40-55) recommends removing positions with >20% missing data prior to analysis, ensuring sufficient data complementarity exists across samples for effective chimeric panel construction while maintaining biological accuracy.

2. The authors state that their approach is aimed at non-model organisms. However, the manuscript would benefit from more explicit contrast with conventional imputation settings-such as human datasets where genotype arrays are used, and external panels are often available. It would be helpful to emphasize that the proposed method is not meant to replace those methods but to extend imputation capabilities where there is no external reference available.

Response: Thank you for this important clarification. We completely agree that our method is not intended to replace conventional imputation approaches where external reference panels are available. We have emphasized throughout the manuscript that, unlike human studies, most organisms lack the resources to assemble functional external panels that enable use of conventional imputation algorithms (page 1, lines 46-52). Therefore, Retriever serves as an alternative for studies lacking external panels to perform imputation for more accurate downstream analyses (page 11, lines 11-28 and lines 49-53).

We have strengthened this positioning in the Discussion section (pages 9-11) by clearly categorizing imputation approaches and positioning Retriever as bridging reference-based and reference-free paradigms rather than competing with well-established human genomics pipelines. Table 1 (page 10) explicitly compares these approaches and clarifies the appropriate application contexts for each methodology.

3. The approach assumes genotype calls are error-free (as stated in the Methods), target samples are genetically similar (i.e., within populations), and high-quality pre-processing has been performed. These assumptions are reasonable but should be more prominently discussed as potential limitations or caveats to consider.

Response: Thank you for this suggestion. We agree these assumptions represent important considerations and have included prominent caveats for users on page 11, lines

54-68. These include: (1) ensuring samples derive from genetically similar populations, as substantial genetic divergence compromises shared haplotype structure assumptions; (2) implementing comprehensive quality control procedures including error-free genotype calling; and (3) appropriate preprocessing including removal of multiallelic sites and positions with excessive missing data.

We emphasize that these considerations are not unique to Retriever but are critical when selecting any imputation approach, as accuracy and reliability depend strongly on underlying data quality and population characteristics. Users must carefully assess dataset suitability and determine whether samples can reasonably be considered as originating from a single population before applying any imputation method.

4. Does the chimeric panel need to be of fixed size across the genome? The algorithm seems to randomly subsample when there are more samples than required. Could it instead use all available data when possible, to improve accuracy or robustness? Would dynamic sizing help?

Response: Thank you for this insightful point. In our current implementation, the chimeric reference panel maintains fixed size across the genome to ensure compatibility with Beagle4's imputation requirements. However, we have considered incorporating additional flexibility by leveraging regions with higher sample counts than required.

One potential enhancement involves prioritizing partitions with greater likelihood of capturing rare alleles and, where feasible, assigning these partitions to the same individual in surrounding buckets to preserve haploblock structure. We are exploring implementation of a ranking system that assigns higher confidence scores to regions with greater sample representation, which could guide imputation by prioritizing more informative regions likely to yield higher accuracy.

While our current approach maintains biological accuracy by preserving local linkage disequilibrium patterns (as demonstrated by consistent >95% accuracy across diverse taxa), we acknowledge that dynamic sizing represents a promising direction for future development that we will consider in subsequent updates.

5. In the comparison between Retriever and external panel-based imputation, the authors remove certain variants from the external reference panel while retaining them in the target samples. This setup inherently favors Retriever and may not reflect realistic use cases. A more informative comparison would involve scenarios grounded in actual differences between sequencing platforms or study designs. The authors should clarify the rationale behind this simulation strategy and discuss its limitations. In particular, it is unclear why, in a real-world setting, a researcher would choose an external reference panel that has lower genomic coverage than the target dataset.

Response: Thank you for drawing attention to this potentially confusing aspect. Our intention with this comparison was to simulate scenarios with incomplete reference coverage where external panels lack variants present in target samples such as outdated reference panels fail to capture recently discovered variant positions present in target samples. As we demonstrate in Figure 4, Beagle cannot impute novel genotypes absent from external reference panels, highlighting a fundamental limitation in handling conditions where missing data depends on observed data but not unobserved data.

This scenario frequently occurs in practice when reference panels constructed from earlier sequencing efforts (often with lower coverage or older technologies) are applied to target samples sequenced with improved technologies. For instance, reference panels built using low-coverage sequencing may systematically miss rare variants that become detectable with high-coverage approaches, creating patterns where missingness depends on technological capabilities rather than biological factors.

This represents a critical real-world limitation where conventional approaches discard potentially valuable population-specific or functionally relevant variants, while Retriever preserves and processes all variants detected in target datasets. We have provided more detailed description of missing data types and their implications on pages 1-2, lines 37-57, and page 6, lines 53-69.

6. The authors report that chromosome 1 takes ~28 hours for chimeric panel construction under high missingness. This raises concerns about scalability to genome-wide analyses. Figure 5 would benefit from more annotation to clarify

whether the times reflect single chromosome runs, and whether parallelization or optimization strategies were considered.

Response: Thank you for this feedback. We reported wall time for chromosome 1 specifically as it represents the largest chromosome and provides a consistent reference point for readers to compare with their own datasets. However, since chimeric assembly requires relatively low computational resources (single CPU core), we strongly recommend running assemblies for all chromosomes in parallel to improve overall efficiency (page 6, lines 98-102).

We have provided code for parallel execution of whole-genome analysis on our GitHub repository. In most cases, processing the entire genome in parallel has minimal impact on overall wall time, as smaller chromosomes require proportionally less time to complete. The primary limitation is the number of available CPU cores for parallel processing rather than the computational demands of individual chromosomes.

Figure 5 indicates these times reflect single chromosome analysis, and we emphasize that genome-wide analysis should be conducted in parallel rather than sequentially to optimize efficiency.

7. In Figure 2, the differences in imputation accuracy across chimeric panel sizes are within ~0.5%, which is small. Are these differences statistically or practically significant? Some additional interpretation would help.

Response: Thank you for raising this important point. While the differences in accuracy across parameter settings appear modest, they represent practical significance due to the sheer number of genotypes analyzed and minor errors carried forward may affect downstream analysis. However, the minor differences also highlight the robustness of our method in constructing effective reference panel analogues.

Nevertheless, we recommend using optimized parameters to achieve highest imputation accuracy. Given the large scale of genomic datasets, even small differences in error rates carry substantial practical significance. For instance, in a dataset of 1 million positions, a 0.5% error rate difference translates to approximately 5,000 incorrectly imputed genotypes, which can be consequential depending on downstream applications.

More importantly, our multivariate analysis (Figure 2C, page 6, lines 2-13) reveals that chimeric panel size explains the majority of variance in imputation accuracy, with optimal performance achieved at 25-50 individuals (12.5-25% of total sample size). This represents a fundamental trade-off in chimeric panel construction where larger panels capture greater genetic diversity but require smaller windows that may disrupt linkage disequilibrium patterns.

8. The sliding window size appears to default to 1kb. How sensitive is imputation accuracy to this choice? Have the authors tested different window sizes or overlap strategies?

Response: Thank you for this question. We tested window sizes ranging from 100 bp to 10 kb and observed that differences in imputation accuracy were negligible, although 1 kb windows yielded the highest performance (Figure 2C, page 6 lines 2-4).

Our multivariate analysis examining both window size and chimeric panel size indicates that chimeric panel size has substantially more impact on accuracy. This likely reflects that when chimeric panels exceed 25% of total sample size, partitions with complete genotypes become very limited, potentially resulting in smaller partitions that disrupt haploblock structures and reduce imputation performance.

The relative insensitivity to window size suggests that the 1 kb default provides an effective balance between preserving local linkage disequilibrium patterns and maintaining sufficient flexibility for regions with varying missing data patterns. We provide detailed discussion of these parameter interactions on pages 4, lines 9-33.

9. The study compares Retriever to external reference panel-based imputation using Beagle4, but lacks comparison with other reference-free methods that were discussed. Some benchmark will be very helpful.

Response: Thank you for this important suggestion. Following recommendations from both reviewers, we conducted comprehensive benchmarking against established reference-free methods including missForest and K-nearest neighbors (KNN) approaches.

We focused on machine learning-based reference-free approaches given their robustness and growing popularity in recent studies. While deep learning methods may offer

sophistication, their high computational demands and longer runtimes present practical limitations, especially for researchers without access to high-performance computing resources or programming expertise.

Our evaluation used human datasets where extensive resources enable reliable benchmarking by directly comparing imputed genotypes to original unmasked genotypes. Results shown in Figure 8 (page 18) and discussed on pages 10, lines 14-24 demonstrate that Retriever + Beagle4 consistently achieved highest accuracy across varying missing data levels, while missForest showed strong potential among reference-free alternatives.

10. The core innovation lies in assembling a chimeric reference panel from complete partitions in the target data. While this is a useful contribution, the imputation itself still relies entirely on Beagle4. The manuscript should better highlight the specific methodological contribution (i.e., reference construction) and clarify that Retriever is a preprocessing framework, not a novel imputation algorithm per se.

Response: Thank you for this clarification. We acknowledge that Retriever is not an imputation algorithm per se and have improved clarity throughout the manuscript to reflect its role as a preprocessing framework. We emphasize on page 11, lines 49-53 that Retriever serves as a preprocessing step that enables conventional imputation algorithms to be applied to datasets lacking external reference panels.

The specific methodological contribution lies in the chimeric reference panel construction methodology that exploits complementary distribution of missing data across samples while preserving essential linkage disequilibrium patterns. This enables established, well-validated imputation algorithms like Beagle4 to be applied in contexts where they were previously inaccessible due to reference panel requirements.

Reviewer #2

Summary: This article proposed Retriever, an approach that builds a "chimeric" reference panel for downstream genomic data imputations by connecting non-overlapping genetic partitions, or windows, where only individuals with non-missing genotype data are selected. As advantages, this approach not only obviates

the requirement of building an external reference panel that needs a large sample size and cost, but also shows unique strength in imputation of novel variants that are absent from previous external reference panels. Furthermore, this approach is especially useful for non-model organisms that lack large-scale genomic data. When compared with reference-free ML approaches that also don't require an external reference panel, it is claimed that Retriever is more biologically interpretable and less complex in implementation.

However, this innovative approach still has limitations, and several comments are listed as follows:

Major Comments

1. In the article, the author compares the proposed approach with both reference-based and reference-free approaches, as shown in Table 1. However, the imputation accuracy (> 95%, >98%, 93-97%) is not fully computed and shown in the result section. During the benchmark, the author has only compared the imputation result between the chimeric and external reference panels. Although the author mentioned that "reference-free approaches demonstrate typically 2-5% lower accuracy compared to reference-based approaches" based on previous studies, no actual numerical results are provided in the article to compare Retriever with any reference-free approaches. Therefore, a more comprehensive benchmark of the Retriever with both reference-based and -free methods using the same training data and test data may be needed to strengthen the current message in Table 1.

Response: Thank you for this excellent suggestion. We have conducted comprehensive benchmarking against established reference-free methods to strengthen our comparative analysis. Specifically, we evaluated our method against widely-used machine learning-based reference-free approaches including missForest and K-nearest neighbors (KNN), given their robustness and growing popularity in recent studies.

While deep learning-based methods may offer greater sophistication, their high computational demands and longer runtimes present practical limitations, particularly for researchers without access to high-performance computing resources or programming

expertise. Therefore, we focused our comparison on accessible, well-established reference-free methods.

We used human datasets for this benchmarking because the extensive resources available for this model organism enable more reliable validation—imputed genotypes can be directly compared to original unmasked genotypes to assess accuracy. Results are presented in Figure 8 (page 18) and discussed pages 10, lines 14-24, demonstrating that Retriever + Beagle4 consistently achieved the highest accuracy across varying levels of missing data, while missForest showed relatively strong performance among reference-free alternatives with maximum iterations set to ten.

2. The paper's imputation results are mostly obtained through the integrated use of Retriever and Beagle4.1 without a genetic map file. From their documents, (https://faculty.washington.edu/browning/beagle/beagle_4.1_21Jan17.pdf), it seems that a constant recombination rate of 1 cM/Mb will be set if no genetic map files are specified. Further clarification/comparison might be needed on how much this will affect the imputation performance when using the Retriever.

Response: Thank you for this important technical point. We conducted additional tests to understand how varying recombination rates affect imputation performance and found minimal impact when using uniform rates (Figure 2B, page 5, lines 103-117). However, we tested only uniform recombination rates applied across genomic positions.

If studies have resources to construct detailed genetic maps, more precise recombination rate information may contribute to higher imputation accuracy. Our current study focuses on Retriever's application to non-model organisms, which often lack resources for processing such detailed recombination information. Therefore, our results represent a conservative scenario—if additional genetic information becomes available, performance should improve rather than degrade.

This represents another advantage of Retriever for non-model organisms: achieving high accuracy (>95%) even under suboptimal conditions where detailed genetic maps are unavailable, while maintaining compatibility with enhanced genetic information when resources permit.

3. Throughout the article, the authors explored and listed several factors (apart from the missing data proportion) that will impact the imputation performance when using Retriever: (1) window size, (2) number of individuals (sample size), and (3) minor allele count (genetic diversity) of the selected population or organism. However, the effect of each factor is measured independently. A potential joint analysis will help users better understand the interaction between these factors and select the optimal choices based on their target data.

Response: Thank you for this valuable suggestion. We conducted multivariate analysis to examine combined effects of key parameters, specifically evaluating interactions between window size and number of chimeric individuals as the most influential factors affecting imputation performance (Figure 2C, page 5, lines 118-121 and page 6 lines 1-17).

Our analysis reveals that chimeric panel size has substantially greater impact on imputation accuracy than window size. When chimeric panels exceed 25% of total sample size, partitions with complete genotypes become limited, potentially creating smaller partitions that disrupt haploblock structures and reduce imputation effectiveness.

Regarding minor allele count, we note this represents a measured variable dependent on the study organism rather than a user-configurable parameter. As with other imputation algorithms, we recommend users consider allele frequency distributions in their datasets before applying the method. Our results indicate that chimeric reference parameters are generally robust, with chimeric panel size being the primary consideration for optimization.

The multivariate analysis provides users with clear guidance: prioritize chimeric panel size optimization (targeting 25% of total samples or 50 individuals, whichever is larger) while maintaining default window sizes, as these parameters show minimal interaction effects.

4. In Materials and methods, the authors noted that preprocessing the input VCF files requires removing multiallelic sites, which is quite common in genomic-based tools. But from the algorithmic perspective (Figure 1), the Retriever seems to be able to handle multiallelic sites. Could Retriever do that with the current implementation? Also, could Retriever handle organisms with polyploidy? It would be great to see the idea from the Retriever used on an even broader scale.

Response: Thank you for this forward-looking suggestion. Our current genotype encoding approach should theoretically accommodate multiallelic sites, but we recommend avoiding such datasets due to complications arising during the decoding process with Beagle4 integration. The complexity emerges not from Retriever's chimeric panel construction but from ensuring compatibility with downstream imputation algorithms.

Polyploidy is not supported in the current implementation, but we recognize this represents an important direction for future development. We will investigate prospects for expanding the algorithm to handle multiallelic and polyploidy datasets in future versions.

These enhancements would indeed broaden Retriever's applicability to agriculturally important crops (wheat, potato, cotton) and other polyploid species that represent significant gaps in current imputation capabilities. We discuss these future directions on page 11, lines 90-107, acknowledging the potential for extending sophisticated genomic analyses to these currently underserved biological systems.

5. In cross-species validation, the author suggests that organisms with higher genetic diversity will result in poorer imputation performance. To represent genetic diversity in the three non-human organisms (*Arabidopsis thaliana*, *Gallus gallus*, and *Saccharomyces cerevisiae*), having a measurable metric like minor allele count would be better, which will help users estimate the imputation accuracy for the data from their organisms.

Response: Thank you for this practical suggestion. We have included minor allele frequency analysis to help users understand expected imputation performance from their organisms (Figure 6B, page 18). This analysis demonstrates the relationship between genetic diversity and imputation accuracy across our test species.

The results confirm expected patterns: *A. thaliana* (derived from inbred lines with relatively homogeneous genetic structure) achieved highest accuracy, while *S. cerevisiae* (representing 8 distinct populations) and *G. gallus* (comprising 13 populations) showed progressively lower but still excellent accuracy correlating with increasing genetic diversity.

These minor allele frequency distributions provide users with benchmarks for estimating likely performance with their own datasets, enabling informed decisions about whether Retriever is appropriate for their specific research contexts.

6. It's great that the GitHub README gives instructions on installing external libraries for the Retriever. Still, a step-by-step tutorial with shell/Python code from preprocessing to output would be more straightforward for users. If possible, adding a tiny data example would be more helpful.

Response: Thank you for this practical suggestion. We completely agree that providing dummy data helps users understand algorithm functionality and verify successful installation. We have added sample datasets and comprehensive scripts demonstrating Retriever workflow beyond the existing GitHub documentation.

The enhanced repository now includes: (1) complete tutorial workflow with shell/Python code from preprocessing through output interpretation, (2) dummy VCF datasets for testing installation and verifying expected performance, (3) parameter optimization examples, and (4) integration scripts for seamless Beagle4 pipeline execution. These additions significantly improve accessibility for users regardless of their computational expertise level.

Minor Comments:

1. The line after the title "Retriever's chimeric framework: Combining methodological strengths" got a typo. Instead of "Retriever strategically bridges reference- and -free and imputation paradigms ...", it should be "Retriever strategically bridges reference-based and -free imputation paradigms ..."

Response: Thank you for catching this error. We have corrected the text as suggested (page 9, line 47).

2. In the Algorithm 1, line 8, the "(-3)" between "missing data" and "occurs" seems confusing. Please correct it if it is a typo.

Response: Thank you for pointing this out. The "(-3)" is not a typo but represents the numerical encoding for missing genotypes in our algorithm. We have included additional

clarification in the algorithm description and supplementary materials (page 13, lines 21-23) to prevent confusion by other readers about this technical implementation detail.

3. The figures are not referenced anywhere in the text.

Response: Thank you for this observation. We have verified that all figures are referenced throughout the manuscript text. The references use "Fig." format rather than "Figure," which may have caused initial confusion. All figure citations are present and appropriately placed within the text to support the corresponding analyses and conclusions.

August 20, 2025

RE: GENETICS-2025-308411

Dear Dr. Zhou:

I am pleased to accept your manuscript titled "Chimeric Reference Panels for Genomic Imputation" for publication in GENETICS, pending minor revision.

Please submit your revision along with a brief description of how you modified the manuscript in response to the reviewers' concerns and suggestions (which can be viewed at the bottom of this email). I expect you should be able to submit a revised manuscript within 30 days. A suitably revised manuscript will be acceptable for publication; I don't expect to send it out for review.

When revising the ms., please make an effort to shorten it, because that almost always improves a manuscript. We urge authors to heed the advice of Strunk and White: "omit needless words"¹. Follow this link to submit the revised manuscript: Link Not Available

Thank you for submitting this story to Genetics.

Sincerely,

Yun Li
Associate Editor
GENETICS

Approved by:
Konrad Lohse
Senior Editor
GENETICS

Reviewer comments:

Reviewer #1 :

Thank the authors for addressing my previous comments and for making the requested edits.

Reviewer #2 :

Summary:

In the revised version, the author improved the clarity of the message that the proposed Retriever is a preprocessing step prior to imputation designed for genetic studies using non-model organisms. The novelty of Retriever still lies in its core idea of building a chimeric reference panel by fully utilizing the limited data for non-model organisms.

To address the previous comments on 1) comparison between Retriever with both reference-based and reference-free methods, 2) the impact of different constant recombination rates, 3) interaction between parameters like sample size and window size, and 4) a quantitative metric for genetic diversity, the author generates new results for each point.

Figure 8 shows that the Retriever + Beagle4 pipeline has better imputation accuracy than the two reference-free methods (KNN and missForest).

Figure 2B shows that the imputation accuracy under different recombination rates is similar, given the same missing data proportion.

Figure 2C shows the imputation accuracy under different sample sizes and window sizes, which seems generally robust.

Figure 6B shows how metrics like minor allele frequency and population structure (number of populations) can significantly affect

the imputation accuracy for the Retriever+Beagle4 pipeline.

Also, in the revised version, the author addressed the previous comment on multiallelic sites and polyploidy in the discussion. The author clarified the implementation limitation from Beagle4 to handle multiallelic sites and regarded both multiallelic sites and polyploidy as important extensions in future versions of Retriever.

Still, one key issue is that Retriever has to rely on Beagle4 to show complete genotype imputation functionality. But with the new clarification added on Page 3 about why Beagle4 is chosen, it seems acceptable to follow this Retriever+Beagle4 design.

Overall, I believe Retriever shows a good, solid impact on the genotype imputation for non-model organisms using its chimeric reference panel.

Minor comments:

1. Both "Accuracy (%)" and "Imputation accuracy (%)" are used as the y-axis label for imputation accuracy in Figure 6, which should be more consistent with other figures.
2. It's great to see that the GitHub page has been improved with more detailed information. The newly added example file "Sample_script.py" is tested and runs smoothly on my Linux server. But still, it would be even better if there were a step-by-step tutorial of the whole Retriever+Beagle4 pipeline, which will be more user-friendly.

Associate Editor comments: Please address reviewer 2's comments carefully.

Dear Professor Yun Li,

Thank you and the reviewers for your constructive feedback and for the opportunity to further improve our manuscript and supplementary materials. Please find below the adjustments made:

1. Fig.6 label consistency

We have rectified the inconsistency in Figure 6 by standardizing the y-axis label. The “Accuracy (%)” has now been revised to use a consistent label (“Imputation accuracy (%)”) that aligns with the terminology used in the other figures, ensuring clarity and uniformity throughout the manuscript.

2. GitHub Repository improvement

We appreciate the positive feedback regarding the improvements to our GitHub repository and are glad to hear that the "Sample_script.py" runs smoothly on reviewer 2's Linux server. In response to your suggestion, we have now added a step-by-step tutorial outlining the complete Retriever+Beagle4 pipeline. This pipeline includes examples for running Retriever and Beagle in one file or across multiple files through parallelization, with the aim of making the workflow more accessible and user-friendly for new users.

3. Manuscript style and clarity

We have also taken care to improve the clarity and conciseness of the manuscript by following the guidance of the Strunk and White, focusing especially on eliminating unnecessary words and improving sentence structure. We have not changed any conceptual, procedural, or descriptive aspects of the paper.

We hope that these revisions address your concerns and enhance the quality and usability of our work. Please let us know if any further modifications are needed.

Best Regards, Meikun Zhou (on behalf of all authors)

September 17, 2025
RE: GENETICS-2025-308411R1

Mr. Meikun Zhou
The University of Queensland
School of the Environment
St Lucia
Brisbane
Australia

Dear Dr. Zhou:

Congratulations, your manuscript titled "Chimeric Reference Panels for Genomic Imputation" is accepted for publication in GENETICS! Many thanks for submitting your research to the journal.

To Proceed to Publication:

1. Format your article according to GENETICS style: <https://academic.oup.com/genetics/pages/author-guidelines>
2. Ensure that you comply with data and community resource citation guidelines: <https://academic.oup.com/genetics/pages/author-guidelines#section-5-9-2>
3. Upload your final files at <https://genetics.msubmit.net>
4. Add oupsupport@scipris.com and genetics.oup@novatechset.com (or the domains @scipris.com and @novatechset.com) to your email program's "safe senders" list. You will be contacted by both at various points during the production process.

Notes:

- Your currently-accepted manuscript (unedited, as submitted, reviewed, and accepted) will be published at GENETICS and deposited into PubMed as an Advance Access article. Notify sourcefiles@thegsajournals.org before signing your license if you do not wish to publish your article via Advance Access.
- We invite you to submit an original color figure related to your paper for consideration as cover art. Please email your submission to the editorial office or upload it with your final files. You can submit a small-sized image for evaluation, and if selected, the final image must be a TIFF file 2513px wide by 3263px high (8.375 by 10.875 inches; resolution of 600ppi). Please avoid graphs and small type.
- After files are sent to Oxford University Press we use SciPris to manage article licensing and payment. If you do not have a SciPris account, you will receive an email from no-reply@scipris.com to sign up to use Oxford University Press' author portal. After logging in, follow the online instructions to sign your license and arrange any payment due.

If you have any questions or encounter any problems while uploading your accepted manuscript files, please email the editorial office at sourcefiles@thegsajournals.org.

Sincerely,

Yun Li
Associate Editor
GENETICS

Approved by:
Konrad Lohse
Senior Editor
GENETICS

Review comments (if applicable):